# pMixFed: Efficient Personalized Federated Learning through Adaptive Layer-Wise Mixup

## Abstract

Partial Personalized Federated Learning (PFL) aims to balance generalization and personalization by decoupling models into shared and personalized layers. However, existing methods typically rely on rigid, static partitioning, which leads to significant global-local model discrepancies, client drift, and catastrophic forgetting. To overcome these limitations, we propose pMixFed, a dynamic, layer-wise PFL approach that integrates an adaptive mixing mechanism (inspired by Mixup) directly into the parameter space. Unlike static methods, pMixFed employs an adaptive strategy to dynamically partition layers and utilizes a gradual transition of personalization degrees to smooth the integration of global and local knowledge. This mechanism effectively mitigates the "hard split" issues found in prior work. Extensive experiments demonstrate that pMixFed consistently outperforms competitive baselines (such as FedAlt and FedSim) in heterogeneous settings, exhibiting faster model training, increased robustness against performance drops, and a self-tuning mechanism that effectively handles cold-start users.

## 1 Introduction

One goal of federated learning (FL) Konečný et al. (2016) is to facilitate collaborative learning of several machine learning (ML) models in a decentralized scheme. FL requires addressing data privacy, catastrophic forgetting, and client drift problem [1] Rostami et al. (2018); Huang et al. (2022); Singhal et al. (2021); Luo et al. (2023); Qu et al. (2022). Existing FL methods cannot address all these challenges with non-Independent and Identically Distributed (non-IID) data. For instance, although Federated Averaging ("FedAvg") McMahan et al. (2017) demonstrates strong generalization performance, it fails to provide personalized solutions for a cohort of clients with non-IID datasets. Hence, the global model, or one "average client" in "FedAvg", may not adequately represent all individual local models in non-IID settings due to client-drift Xiao et al. (2020). Personalized FL (PFL) methods handle data heterogeneity by considering both generalization and personalization during the training stage. Since there is a trade-off between generalization and personalization in heterogeneous environments, PFL methods leverage heterogeneity and diversity as advantages rather than adversities Pye & Yu (2021); Tan et al. (2022). A group of PFL approaches train personalized local models on each device while collaborating toward a shared global model. Partial PFL, also known as parameter decoupling, involves using a partial model sharing, where only a subset of the model is shared while other parameters remain "frozen" to balance generalization and personalization until the subsequent round of local training.

While partial PFL methods are effective in mitigating catastrophic forgetting, strengthening privacy, and reducing computation and communication overhead Pillutla et al. (2022); Sun et al. (2023), there are still some unaddressed challenges. First, the question of *when, where, and how to optimally partition the full model*, remains unresolved. Recent studies Pillutla et al. (2022); Sun et al. (2023) have shown that there is no "one-size-fits-all" solution; the best or optimal partitioning strategy depends on factors such as task type (e.g., next word prediction or speech recognition) and local model architecture. An improper partitioning choice

---

[1] A phenomenon where the global model fails to serve as an accurate representation because local models gradually drift apart due to high data heterogeneity.

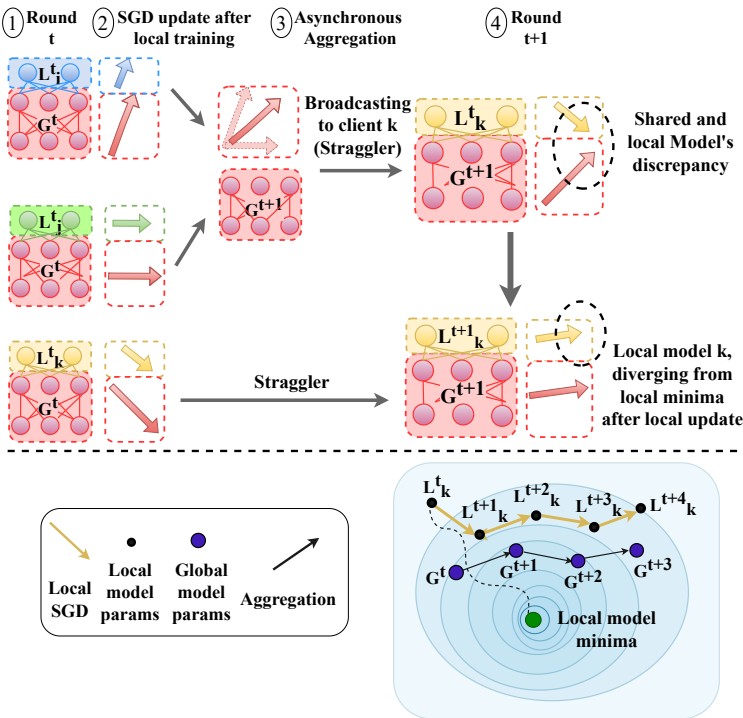

Figure 1: Discrepancy between personalized and global shared layers in Partial PFL: **(1)** The global model, $G^t$, is constructed by aggregating asynchronous local updates from clients, denoted as $L_i^t$, $L_j^t$, and $L_k^t$. **(2),(3)** In communication round $t$, available clients $i$ and $j$ aggregate shared parameters to produce the updated global model $G^{t+1}$, while the personalized parameters, such as $L_k^t$, remain unchanged for unavailable clients. **(4)** This integration of distinct models, $G^{t+1}$ and $L_k^t$, induces inconsistencies in the overall model updates. **(Bottom)** During the joint training of generalized and personalized models, the gradient updates from the generalized layers are impacted by the gradients from personalized layers, resulting in catastrophic forgetting, performance drop and slower convergence rates.

can lead to issues such as underfitting, overfitting, increased bias, and catastrophic forgetting. Some studies Liang et al. (2020) suggest that personalized layers should reside in the base layers, while others Collins et al. (2021); Arivazhagan et al. (2019) argue that the base layers contain more generalized information and should be shared. Further, the use of a fixed partitioning strategy across all communication rounds for heterogeneous clients can limit the efficacy of collaborative learning. For instance, if the performance of the client suddenly drops due to new incoming data, the partitioning strategy should be changed because the client requires more frozen layers. Another issue is catastrophic forgetting of the previously shared global knowledge after only a few rounds of local training because the shared global model can be completely overwritten by local updates leading to generalization degradation Luo et al. (2023); Shirvani-Mahdavi et al. (2023); Huang et al. (2022); Xu et al. (2022). Most importantly, partial models may experience slower convergence compared to full model personalization, as frozen local model updates can diverge in an opposite direction from the globally-shared model. Since the generalized and personalized models are trained on non-IID datasets, there might also be a domain shift, leading to model discrepancy as depicted in Figure 1. These discrepancies arise from variations in local and global objective functions, differences in initialization, and asynchronous updates Yang et al. (2024); Lee et al. (2023b). As a result, merging the shared and the personalized layers can disrupt information flow within the network, impede the learning process, and lead to a slower convergence rate or accuracy drop in partial PFL models such as FedAlt and FedSim Pillutla et al. (2022)[2]. Further, while partial PFL techniques contribute to an overall improved training accuracy, they can reduce the test accuracy on some devices, particularly in devices with limited samples, leading to variations in results in terms of the performance level Pillutla et al. (2022). Hence, there is a need for novel solutions to achieve the following goals in PFL:

---

[2]More details on this is discussed in section 5.3.

- Dynamic and Adaptive Partitioning: The balance between shared and personalized layers should be dynamically and adaptively adjusted for each client during every communication round, rather than relying on a static, fixed partitioning strategy for all participants.

- Gradual Personalization Transition: The degree of personalization should transition gradually across layers, as opposed to an "all-or-nothing" approach that employs strict partitioning or hard splits within the model discussed in Figure 1. This ability allows nuanced adaptation for individual client needs due to heterogeneity.

- Improved Generalization Across All Clients: The average personalization accuracy should be such that the global model is unbiased toward specific subsets of clients.

- Mitigation of Catastrophic Forgetting: The strategy should address the catastrophic forgetting problem by incorporating mechanisms to strengthen the generalization and retain the state of the previous global model when updating the global model in aggregation.

- Scalability and Adaptability: The approach should be fast, scalable, and easily adaptable to new cold-start clients while accounting for model/device heterogeneity.

To achieve the above, we propose "pMixFed", a layer-wise, dynamic PFL approach that integrates Mixup Zhang et al. (2017) between the shared global and personalized local models' layers during both the broadcasting (global model sharing with local clients) and aggregation (aggregating distributed local models to update the global model) stages within a partial PFL framework. While Mixup is traditionally a data augmentation technique, we mathematically adapt its interpolation principles to the parameter space to solve the discrepancy problem discussed in Figure 1. Our main contributions include:

- We develop an online, dynamic interpolation method between local and global models using Mixup Yoon et al. (2021), effectively addressing data heterogeneity and scalable across varying cohort sizes, degrees of data heterogeneity, and diverse model sizes and architectures.

- Our solutions facilitate a gradual increase in the degree of personalization across layers, rather than relying on a strict cut-off layer, helping to mitigate client drift.

- We introduce a new fast and efficient aggregation technique which addresses catastrophic forgetting by keeping the previous global model state.

- "pMixFed" reduces the participant gap (test accuracy for cold-start users) and the out-of-sample gap (test accuracy on unseen data) caused by data heterogeneity through linear interpolation between client updates, thereby mitigating the impact of client drift.

The remainder of this paper is organized as follows. In Section 2, we review the relevant literature on personalized federated learning, highlighting existing methodologies and their limitations. Section 3 introduces the problem formulation, including both fully personalized and partially personalized federated learning settings, accompanied by theoretical analysis. In Section 4, we present the proposed *pMixFed* framework, detailing the integration of adaptive and dynamic mixup strategies across different model layers. Section 5 provides extensive empirical evaluations of *pMixFed*, including comparisons with state-of-the-art baselines and analyses of computational and communication efficiency. Finally, Section 6 concludes the paper and outlines potential directions for future research.

## 2   Related Work

PFL seeks to adapt each client's model to its individual data, preferences, and context. Similar to classic domain adaptation, data privacy can be an important concern in PFL Stan & Rostami (2024); Verma et al. (2025); Bingtao et al. (2025); Tian et al. (2024), the additional challenge, however, is that data is distributed and we need to adapt several models due to data heterogeneity between the clients. Since the advent of FL, a wide range of PFL methods have been proposed to address client heterogeneity (both statistical and

system), which we group into four categories: (i) data-centric strategies; (ii) clustering-based approaches; (iii) global-model adaptation; and (iv) local-model personalization—the focus of this work, which also encompasses partial-model (parameter-decoupling) techniques. These approaches have been discussed in more detail in the Appendix A, and Partial PFL models which are a subcategory of the local model personalization, have been discussed below.

A prominent subcategory of local-model personalization, which decouples a shared parameter set (e.g., backbone) from client-specific modules (e.g., heads/adapters), is called **partial model personalization**. This design is widely adopted because it is parameter-efficient, easy to quantize or deploy, and operationally transparent. By keeping the shared component stable and personalizing only a small subset, these methods reduce negative transfer across clients and mitigate catastrophic forgetting of global knowledge while enabling fast on-device adaptation. FedPer Arivazhagan et al. (2019) introduced partial models in FL, sharing only initial layers with generalized information and reserving final layers for personalization. FedBABU Oh et al. (2021) divides the network into a shared body and a frozen head with fully connected layers. Other frameworks, such as FURL Bui et al. (2019) and LG-FedAvg Liang et al. (2020), apply partial PFL by retaining private feature embeddings or using compact representation learning for high-level features, respectively. Two baseline methods in this paper are FedAlt Singhal et al. (2021) and FedSim Pillutla et al. (2022). FedAlt uses a stateless FL paradigm, reconstructing local models from the global model, while FedSim synchronously updates shared and local models with each iteration. FedSelect Tamirisa et al. (2024) is a Lottery-ticket-inspired subnetwork selection method that has been published recently. This method gradually personalizes a subnetwork per client while aggregating the rest. FedTSDP Zhu et al. (2024) is a PFL method that combines partial-model with clustering to better handle non-IID data. FDLoRA Qi et al. (2024), which is an LLM-based FL systems, use LoRA (Low-Rank Adaptation) technique to improve personalization. FDLoRA introduces dual LoRA modules per client, one capturing global knowledge aggregated by the server, and the other capturing client-specific personalization. An adaptive fusion process reconciles the two, effectively balancing generalization with personalization without requiring full model sharing.

These methods face challenges such as model update discrepancies (as shown in Figure 1) and catastrophic forgetting, where shared layers may undergo significant changes after only a few rounds of local training, resulting in sudden accuracy drops especially in high-scale Fl setting. Additionally, users with high personalization accuracy may freeze more layers than cold-start users or unreliable participants, who should rely more heavily on the global model. These challenges have motivated our development of a dynamic, adaptive layer-wise approach to balance generalization and personalization across clients, allowing tuning at different communication rounds to accommodate varying performance conditions.

## 3  Problem Formulation

Consider $K$ collaborating clients (or agents), each trying to optimize a local loss function $F_k(\theta)$ on the distributed local dataset $D_k = (x_k, y_k)$, where $(x, y)$ shows the data features and the corresponding labels, respectively. Since the agents collaborate, the parameters $\theta$ (parameters of the global model) are shared across the agents. A basic FL objective function aims to optimize the overall global loss:

$$\min_{\theta} F(\theta) = \sum_{k=1}^{K} \frac{|\mathcal{D}_k|}{|\mathcal{D}|} F_k(\theta) \qquad , \tag{1}$$

$$F_k(\theta) = \frac{1}{|\mathcal{D}_k|} \sum_{(x, y_i) \in \mathcal{D}_k} \ell(f_k(\theta, x_i), y_i)$$

where $|\mathcal{D}| = \sum_{k=1}^{K} |\mathcal{D}_k|$ and $F(\theta)$ is the global loss function of *FedAvg* McMahan et al. (2017). FL is performed in at iterative fashion. At each round, each client downloads the current version of the global model and trains it using their local data. Clients then send the updated model parameters to the central server. The central server uses the model updates from the selected clients and aggregates them to update the global model. Iterations continue until convergence.

In Equation 1, the assumption is that the data is collected from an IID distribution, and all clients should train their data according to the exact same model. However, this assumption cannot be applied within many practical FL settings due to the non-IID nature of data and resource limitations Imteaj et al. (2021);

Imteaj & Amini (2021). In the PFL settings with high heterogeneity and non-IID data distribution, the same issue persists and the local parameters need to be customized toward each agent. PFL extends FL by solving the following Li et al. (2021a); Pillutla et al. (2022):

$$\min_{\theta, \theta_k} \sum_{k=1}^{K} \frac{1}{|\mathcal{D}_k|} (\mathcal{F}_k(\theta_k) + \alpha_k \|\theta_{\mathbf{k}} - \theta\|^2). \tag{2}$$

PFL explicitly handles data heterogeneity through the term $\mathcal{F}_k(\theta_k)$ which accounts for model heterogeneity by considering personalized parameter $\theta_k$ for client $k$. Meanwhile, $\theta$ represents the shared global model parameters in Equation 2, and $\alpha_k$ acts as a regularizer and indicates the degree of personalization tuning collaborative learning between personalized local models $\theta_k$ and generalized global model $\theta$. When $\alpha_k$ is small, the personalization power of the local models will be increased. If $\alpha_k$ is large, the local models' parameters tend to be closer to the global parameters $\theta$.

### 3.1 Partial Personalized Model

The limitations of full model personalization methods with global and fully independent local models are discussed in Section 2. Partial PFL methods improve personalization by providing more flexibility through allowing clients to choose which parts of their models to be personalized based on their specific needs and constraints for improved performance. Let $L_k^t$ be a partial local model $k$ in round $t$ which is partitioned into two parts $\langle L_{l,k}^t; L_{g,k}^t \rangle$, where $l, g \subseteq \{1, \ldots M\}$ are the personalized and global layers, and $M$ is the number of layers. We can integrate both personalized and generalized layers in a local model $L_k$ as:

$$\mathcal{F}_k(\theta_k) = \ell(f_k(\langle L_{g,k}; L_{l,k} \rangle, x_k), y_k) \tag{3}$$

Among different partitioning strategies for partial PFL Pillutla et al. (2022), the most popular technique is to assign local personalized layers $L_{l,k}^t$ to final layers and allow the base layers $L_{g,k}^t$ to share the knowledge similar to *FedPer* Arivazhagan et al. (2019). This choice aligns with insights from MAML [3] algorithm, suggesting that initial layers keeps general and broad information while personalized characteristics manifest prominently in the higher layers. Accordingly, we would have:

$$\mathcal{F}_k(\theta_k) = L_l^{(t)}(L_g^{(t)}(x_k)) \xrightarrow{localupdate} L_l'(L_g'(x_k)) \quad ,$$

$$L_l'(L_g'(x_k)) \xrightarrow{broadcasting} L_l^{(t+1)}(G^{(t+1)}(x_k))$$

For simplicity $L_g = L_{g,k}$ and $L_l = L_{g,k}$ where $\{1 \leq g \leq s \leq l \leq M\}$ and $s$ is the split(cut) layer. The objective in solving Equation 3.1 is to find the optimal $s$ (cut layer) which minimizes the personalization objective: $\sum_{k=1}^{K} \frac{1}{|\mathcal{D}_k|} \mathcal{F}_k(\theta_k) = L_l^{(t)}(L_g^{(t)}(x_k))$. In partial models, after several rounds of local training, both personalized and global layers of local model are updated. This update could be synchronous like *FedSim* or asynchronous as in *FedAlt*. The Personalized layers will be frozen until the next communication round, $L_l^{(t+1)} = L_l'$ and the global layers will be sent to the server for global model aggregation : $G^{(t+1)} \leftarrow \sum_{k=1}^{K} \frac{|\mathcal{D}_k|}{|\mathcal{D}|} L_g'(x_k)$. In the next broadcasting phase, the shared layers of the local model will be updated as $L_g^{(t+1)} \leftarrow G^{(t+1)}$. For details of the theoretical analysis, see the appendix section C.

## 4 Methodology

In this section, we present our FL approach, which incorporates layer-wise *Mixup* in the feature space. We begin by briefly defining the *Mixup* technique, followed by a detailed description of the proposed algorithm and its integration into the federated setting. We also discuss the rationale behind this design and explain how it facilitates improved personalization across clients.

---

[3]Model-Agnostic Meta-Learning

### 4.1 Mixup

Mixup is a data augmentation technique for enhancing model generalization Zhang et al. (2017) based on learning to generalize on linear combinations of training examples. Variations of Mixup have consistently excelled in vision tasks, contributing to improved robustness, generalization, and adversarial privacy. Mixup creates augmented samples as:

$$\bar{x} = \lambda.x_i + (1 - \lambda).x_j$$
$$\bar{y} = \lambda.y_i + (1 - \lambda).y_j, \tag{4}$$

where $x_i$ and $x_j$ are two input samples, $y_i$ and $y_j$ are the corresponding labels, $\lambda$, $\lambda \sim \text{Beta}(\alpha, \alpha), \lambda \in [0, 1]$, is the degree of interpolation between the two samples. Mixup relates data points belonging to different classes which has been shown to be successful in mitigating overfitting and improving model generalization Verma et al. (2019); Guo et al. (2019); Zhang et al. (2020). There are many variants of Mixup that has been developed to address specific challenges and enhance its effectiveness. For instance, AlignMixup Venkataramanan et al. (2022) improves local spatial alignment by introducing transformations that better preserve semantic consistency between input pairs. Manifold Mixup Verma et al. (2019) extends the concept to hidden layer representations, acting as a powerful regularization technique by training deep neural networks (DNNs) on linear combinations of intermediate features. CutMix Yun et al. (2019) replaces patches between images, blending visual information while retaining spatial structure. Remix Chou et al. (2020) addresses class imbalance by assigning higher weights to minority classes during the mixing process, enhancing the robustness of the trained model on imbalanced datasets. AdaMix Guo et al. (2019) dynamically optimizes the mixing distributions, reducing overlaps and improving training efficiency. This linear interpolation also serves as a regularization technique that shapes smoother decision boundaries, thereby enhancing the ability of a trained model to generalize to unseen data. Mixup also increases robustness against adversarial attacks Zhang et al. (2020); Beckham et al. (2019); and improves performance against noise, corrupted labels, and uncertainty as it relaxes the dependency on specific information Guo et al. (2019).

### 4.2 pMixFed : Partial Mixed up Personalized federated learning

Our goal is to leverage the well-established benefits of Mixup in the context of personalized FL. While Mixup has previously been employed in FL frameworks, such as XORMixup Shin et al. (2020), FEDMIX Yoon et al. (2021), and FedMix Wicaksana et al. (2022), prior studies have primarily focused on using Mixup for data augmentation or data averaging. We propose **pMixFed** by integrating Mixup on the model parameter space, rather than using it on the feature space. Mathematically, this operates as a layer-wise convex combination (linear interpolation) of the model weights, adapting the data-space Mixup principle to the parameter space.The Mixup has been applied between the parameters of the global and the local models in a layer-wise manner and eliminates the need for static and rigid partitioning strategies. Specifically, during both the broadcasting and aggregation stages of our partial PFL framework, we generate mixed model weights using an interpolation strategy which is illustrated in Figure 2. Mixup offers the flexibility in combining models by introducing a mix degree for each layer $\lambda_i$, which changes gradually according to $\mu$, i.e., the *mix factor*. Parameter $\mu$ is also updated adaptively in each communication round and for each client according to the test accuracy of the global and the local model during the evaluation phase of FL. $\mu$ is computed as follows:

$$\mu_k^t = 1 - \frac{1}{1 + e^{-\delta(Acc^b - 1)}} \tag{5}$$

$\delta = (t/T)$, where $t$, and $T$ are the current communication round, and the overall number of communication rounds respectively. Also, $Acc$ calculated as: $Acc = (Acc_k^t / Acc_{overall}^{(t-1)})$ in the *broadcasting* phase and average test accuracy of the previous global model $(Acc = G^t(x, \theta^t), y)$ on all local test sets $x = \{x_1, x_2, \ldots, x_K\}$. , in the *aggregation* stage. $b$ is the offset parameter for sigmoid function which is set to 2 after several experiments. More detailed discussion on parameter $\mu$'s rule of update is discussed in section 5.4 and the reason behind this formulation is discussed in the Appendix Sec. 2.

As shown in Figure 2, Mixup is applied in two distinct stages of FL. Firstly, when transferring shared knowledge to local models, the local model $L_k$ is mixed up with the current global model $G$ according

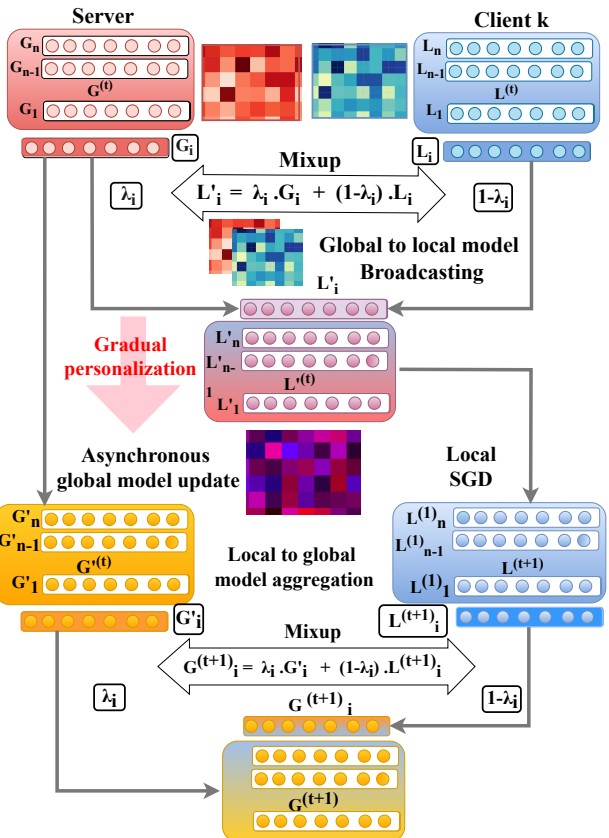

Figure 2: Workflow of pMixFed: Mixup is used in two stages. **1-Broadcasting:** when transferring knowledge to local models, the frozen personalized model $L_k^{(t)}$ is mixed up with global model $G^{(t)}$ according to the adaptive mix factor $\mu_k^{(t)}$ which determines layer-wise mixup degree $\lambda_i$ for layer $i$. **2-Aggregation:** The updated global model $G^{(t+1)}$ is generated through applying Mixup between the updated local model $L^{(t+1)}$ and the current global model $G'^{(t)}$ state.

to the dynamic mixing factor $\mu$, which determines the change ratio of $\lambda_i$ (layer-wise Mixup degree in Eq. equation 4) across different layers. $\lambda_i$ gradually is changed from $1 \rightarrow 0$ as we move from the head to the base layer. $\lambda_i = 1$ means sharing the 100% of the global model and $\lambda_i = 0$ means that the corresponding layer in local model is frozen and will not be mixed up with the global model $G$. Calculation of the Mixup degree of layer $i$ $\lambda_i$ at both broadcasting and aggregation stages is performed as follows:

$$
\begin{aligned}
\text{Broadcasting Stage:} \quad \lambda_i &= \begin{cases} 1 & \lambda_i > 1 \\ \mu * (n - i) & \lambda_i \leq 1 \end{cases} \\
\text{Aggregation Stage:} \quad \lambda_i &= \begin{cases} 0 & \lambda_i \leq 0 \\ 1 - (i * \mu) & \lambda_i > 0, \end{cases}
\end{aligned}
\tag{6}
$$

where $n$ is total number of layers, $i$ is the current layer number starting from the base layer to the head as $i = 0 \rightarrow i = n$. $\mu$ is the mix factor which will be adaptively updated in each communication round $t$ and for each local model $k$ according to Eq. 5.

### 4.2.1   Broadcasting : Global to local model transfer

This stage involves sharing global knowledge with local clients. In the existing PFL methods, the same weight allocation is typically applied to each heterogeneous local model. In our work, we personalize this process by allowing the local model to select the proportion of layers it requires. For instance, for a cold-start user,

more information should be extracted from the shared knowledge model, implying that a few layers should be frozen for personalization. Accordingly, A history of the previous global model $G^t$ will remain which helps with catastrophic forgetting of the generalized model. Additionally, we introduce a gradual update procedure where the value of $\lambda$ gradually decreases from one (indicating fully shared layers) from the base layer to the end, based on the mixing factor $\mu$. The mix layer is adaptively updated in each communication round for each client individually, according to personalization accuracy. With this adaptive and flexible approach, not only can upcoming streaming unseen data be managed, but also the participation gap (test accuracy of new cold start users) would be improved. The Broadcasting phase is illustrated in Algorithm 1 which is inspired by Kairouz et al. (2021). The update rule of local model $L_k^{(t)}$ is as follows:

$$
\begin{aligned}
L'^{(t)}_{k,i} &= (\lambda^{(t)}_{k,i}).G_i^{(t)} + (1 - \lambda^{(t)}_{k,i}).L_{k,i}^{(t)} \\
L_k^{(t+1)} &= L'^{(t)}_k - \eta \nabla F_k(L'^{(t)}_k).
\end{aligned}
\tag{7}
$$

### 4.2.2 Aggregation: Local to Global Model Transfer

Existing methods primarily categorize layers into two types: personalized layers and generalized layers. The updated global layers from different clients, $G_k^{(t+1)}$, are typically aggregated using Equation 3, which can lead to catastrophic forgetting. Since the base layers of the global model serve as the backbone of shared knowledge Raghu et al. (2019), This issue arises because, during each local update, the generalized layers undergo substantial modifications Pillutla et al. (2022); Oh et al. (2021). When the global model is updated by simply aggregating local models, valuable information from previously shared knowledge may be lost, leading to forgetting—even if $G^t$ performs better than the newly aggregated model. To address this challenge, we propose a new strategy that applies Mixup between the gradients of the previous global model and each client's local model before aggregation. For each client $i$, the mixup coefficient $\lambda_i$ gradually increases from 0 to 1, moving from the head to the base layer, controlled by the mix factor $\mu_i$. Additionally, the base layer is adaptively updated based on the communication round and the generalization accuracy, ensuring robust integration of shared and personalized knowledge. It should be noted that parameter $\mu$ is constant in the aggregation stage for all clients as it's dependent to the average performance of the previous global model.

$$
\begin{aligned}
G'^{(t+1)}_{k,i} &= (\lambda^{(t)}_{i,k}).G_i^{(t)} + (1 - \lambda^{(t)}_{k,i}).L_{k,i}^{(t+1)}, \\
G^{(t+1)} &= \sum_{k=1}^{K} \frac{|D_k|}{\sum_{k=1}^{K} |D_k|} G'^{(t+1)}_k.
\end{aligned}
\tag{8}
$$

The high-level block-diagram visualization of the proposed method is shown in Figure 2. It is important to note that the sizes of local models $LM_i^{(0)}$ can differ from each other. Consequently, the size of $GM^{(0)}$ should be greater than the maximum size of local models. The parameter $\lambda$ in Equation 4 determines the Mixup degree between the shared model $GM$ and the local models $LM_i^{(\cdot)}$, while $\mu$ governs the slope of the change in $\lambda$ across different layers. The degree of Mixup gradually decreases according to the parameter $\mu$ from 0 to 1. In this scenario, $\lambda = 0$ for the first base layer, indicating total sharing, while $\lambda = 1$ applies to the final layer, which represents no sharing. The underlying concept is that the base layer contains more general information, whereas the final layers retain client-specific information. The use of the parameter $\mu$, relative to the number of local layers, eliminates the need for a specified cut layer $k$ and allows its application across different model sizes and layers. The parameter $\mu_i$ is adaptively updated based on the personalized and global model accuracy for each client. Algorithm 1 shows how Mixup is used as a shared aggregation technique between individual clients and the server. In each training round, only one client is *Mixed up* with the global model, and $\lambda$ is adaptively learned based on the objective function using online learning. Algorithm 2 shows how Mixup is employed as a shared aggregation technique between the clients and the server. In each training round, only one client is "Mixed up" with the global model and the $\lambda$ parameter is adaptively learned based on the objective function using online learning.

**Algorithm 1** Broadcasting: global to local model transfer

1: **Input:** Initial states global model: $G^{(0)}$, local models: $\{L_i^{(0)}\}_{i=1,\dots,N}$, Number of communication rounds $T$, Number of communication rounds $T$, Number of local training rounds $Itr$, Number of layers in local models $\{L_i\}$
2: **for** $t = 0, 1, \dots, T-1$ **do**
3:     Server selects $K$ devices $S(t) \subset \{1, \dots, N\}$
4:     Update $\mu$ for each $L_i$, $i = \{1, \dots, K\}$
5:     Server broadcasts $G^{(t)}$ to each device in $S(t)$
6:     **for** each device $k \in S(t)$ **do**
7:         $L_k'^{(t)} = \text{Mix}[(L_k^{(t)}, G_k^{(t)}), \mu_{Broad}]$
8:         **for** $l = 0, 1, \dots, Itr - 1$ **do**
9:             $L_k'^{(t+1)} \xleftarrow{Localtrain} (L_k'^{(t)}, \eta)$
10:         **end for**
11:         Each device saves the updated model $L_k^{(t+1)} = L_k'^{(t+1)}$
12:     **end for**
13: **end for**

**Algorithm 2** Aggregation:local to global model transfer

1: **Input:** Initial states global model: $G^{(0)}$, Number of communication rounds $T$, number of devices per round $s(t)$, Mix factor $\mu$
2: **for** $t = 0, 1, \dots, T-1$ **do**
3:     **for** each device $k \in S(t)$ **do**
4:         Update $\mu_k^t$ for $k$ in round $t$
5:         $G_k'^{(t+1)} = \text{Mix}[(L_k^{(t+1)}, G_k^{(t)}), \mu_{Agg}]$
6:     **end for**
7:     Each Device sends $G_k'^{(t+1)}$ back to server
8:     $G^{(t+1)} \xleftarrow{Agg} G_k'^{(t+1)}{}_{k \in S(t)}$
9: **end for**

## 5 Experiments

### 5.1 Experimental Setup

#### 5.1.1 Datasets:

We used three datasets widely used in federated learning: **MNIST** LeCun et al. (1998), **CIFAR-10**, **CIFAR-100** Alex (2009), and **Caltech-101**. CIFAR-10 consists of 50,000 images of size $32 \times 32$ for training and 10,000 images for testing. CIFAR-100, on the other hand, consists of 100 classes, with 500 $32 \times 32$ images per class for training and 100 images per class for testing. MNIST contains 10 labels and includes 60,000 samples of $28 \times 28$ grayscale images for training and 10,000 for testing. Caltech-101 is a canonical benchmark for object recognition, introduced by Li and colleagues in 2003. The dataset comprises 9,146 images distributed across 101 object categories—such as helicopter, elephant, and chair—plus a background/clutter class. Category sizes range from roughly 40 to 800 images (with most near 50), and images are typically around $300 \times 200$ pixels.

We followed the setup in McMahan et al. (2017) and Oh et al. (2021) to simulate heterogeneous, non-IID data distributions across clients for both train and test datasets. The maximum number of classes per user is set to $S = 5$ for CIFAR-10 and MNIST, and $S = 50$ for CIFAR-100. Experiments were conducted across varying heterogeneous settings, including small-scale ($N = 10$) and large-scale ($N = 100$) client populations, with different client participation rates $C = [100\%, 10\%]$ to measure the effects of stragglers. For Caltech-101 experiment setup, please refer to the appendix E.3.

#### 5.1.2 Training Details:

For evaluation, we have reported the average test accuracy of the global model Yuan et al. (2021) for different approaches. The final global model at the last communication round is saved and used during the evaluation. The global model is then personalized according to each baseline's personalization or fine-tuning algorithm for $r = 4$ local epochs and $T = 50$. For **FedAlt**, the local model is reconstructed from the global model and fine-tuned on the test data. For **FedSim**, both the global and local models are fine-tuned partially but simultaneously. In the case of **FedBABU**, the head (fully connected layers) remains frozen during local training, while the body is updated. Since we could not directly apply **pFedHN** in our platform setting,

we adapted their method using the same hyper parameters discussed above and employed hidden layers with 100 units for the hypernetwork and 16 kernels. The local update process for **LG-FedAvg**, **FedAvg**, and **Per-FedAvg** simply involves updating all layers jointly during the fine-tuning process. The global learning rate for the methods that need sgd update in the global server e.g., FedAvg, has been set from $lr_{global} = [1e-3, 1e-4, and 1e-5]$. It should be noted that due to the performance drop for some methods (FedAlt , FedSim) in round 10 or 40 in some settings, we've reported the highest accuracy achieved. Also this is the reason the accuracy curves are illustrated for 39 rounds instead of 50. [4]

### 5.1.3 Baselines and backbone:

We compare our method **pMixFed**: an adaptive and dynamic mixup-based PFL approach, against several baselines. These baselines include: **FedAvg** McMahan et al. (2017), **FedAlt** Pillutla et al. (2022), **FedSim** Pillutla et al. (2022), **FedBABU** Oh et al. (2021). Additionally, we compare against full model personalization methods, **Ditto** Li et al. (2021b), **FedRep** Yang et al. (2023) **Per-FedAvg** Fallah et al. (2020) , and **LG-FedAvg** Liang et al. (2020). For all experiments, the number of local training epochs was set to $r = 4$, number of communication rounds was fixed at $T = 50$, and the batch size was 32. The Adam optimizer has been used while the learning rate, for both global and local updates, was $lr = 0.001$ across all communication rounds. The average personalized test accuracy across individual client's data in the final communication round is reported in Table 2 and 1. Figure 3 presents the training accuracy versus communication rounds for CIFAR10 and CIFAR100 datasets.

### 5.1.4 Model Architectures:

Following the FL literature, we utilized several model architectures. For MNIST, we used a simple CNN consisting of 2 convolutional layers (each with 1 block) and 2 fully connected layers. for CIFAR10 and CIFAR100, we employed a CNN with 4 convolutional layers (1 block each) and 1 fully connected layer for all datasets. Additionally, we used MobileNet, which comprises 14 convolutional layers (2 blocks each) and 1 fully connected layer, for CIFAR-10 and CIFAR-100. For partial model approaches such as **FedAlt** and **FedSim**, the split layer is fixed in the middle of the network: for CNNs, layers [1,2] are shared, while for MobileNet, layers [1–7] are considered as shared part. More details about the training process are discussed in Appendix Sec. 4.1. Experimental Setup. Our implementation is also available as a supplement for reproducing the results.

Table 1: Average Test Accuracy of different methods using Mobile-Net model. See more details in section 5.2

| Method | CIFAR-10 | | | | CIFAR-100 | | | |
| | $N = 100$ | | $N = 10$ | | $N = 100$ | | $N = 10$ | |
| | $C = 10\%$ | $C = 100\%$ | $C = 10\%$ | $C = 100\%$ | $C = 10\%$ | $C = 100\%$ | $C = 10\%$ | $C = 100\%$ |
|---|---|---|---|---|---|---|---|---|
| FedAvg | 26.93 | 25.61 | 15.64 | 19.55 | 4.50 | 4.54 | 17.65 | 21.24 |
| FedAlt | 39.36 | 46.06 | 39.60 | 42.79 | 27.85 | 19.07 | 48.30 | 14.31 |
| FedSim | 51.12 | 45.43 | 40.30 | 35.43 | 28.18 | 19.52 | 47.41 | 48.30 |
| FedBaBU | 28.70 | 25.98 | 17.58 | 20.74 | 4.63 | 4.72 | 17.46 | 17.57 |
| Ditto | 26.58 | 62.77 | 18.60 | 43.98 | 7.00 | 16.07 | 16.89 | 20.74 |
| FedRep | 29.60 | 31.25 | 32.46 | 51.73 | 14.23 | 28.03 | 33.75 | 45.76 |
| Per-FedAvg | 34.30 | 43.59 | 19.52 | 38.29 | 24.04 | 30.65 | 16.27 | 37.20 |
| Lg-FedAvg | 34.52 | 34.17 | 48.88 | 58.51 | 5.65 | 5.73 | 31.89 | 35.78 |
| **pMixFed** | **69.94** | **72.42** | **54.90** | **74.62** | **45.62** | **56.63** | **54.71** | **58.25** |

## 5.2 Comparative Results

We evaluated our proposed method: **pMixFed**, which utilizes both adaptive and dynamic layer-wise mixup degree updates, and **pMixFed-Dynamic**, where the mixup coefficient $\mu_i$ is fixed across all training rounds and gradually decreases from $1 \rightarrow 0$ from the head to the base layer ($i = M \rightarrow 1$) following a Sigmoid

---

[4]Change of hyper-parameters such as lr, batch size, momentum and even changing the optimizer to Adam did not mitigate the performance drop in most cases.

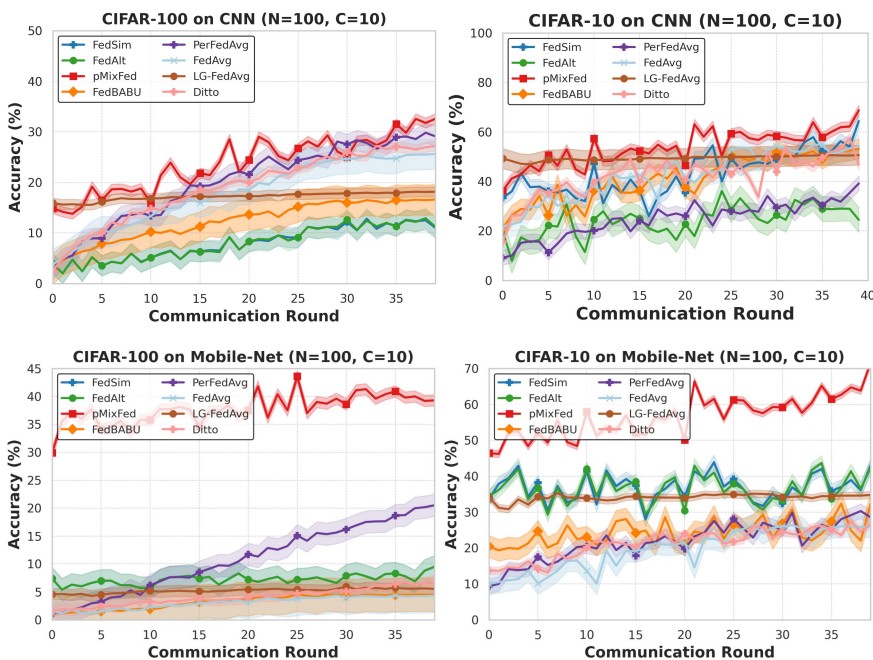

Figure 3: Average Test accuracy in different global communication rounds for *pMixFed* and other PFL baselines experimented on CIFAR10 and CIFAR100 where N=100, C=10%. More details are discussed in Section 5.3

Table 2: Average Test Accuracy of different methods using CNN model. See more details in section 5.2

| Method | CIFAR-10 | | | | CIFAR-100 | | | | MNIST | |
|---|---|---|---|---|---|---|---|---|---|---|
| | $N = 100$ | | $N = 10$ | | $N = 100$ | | $N = 10$ | | $N = 100$ | $N = 10$ |
| | $C = 10\%$ | $C = 100\%$ | $C = 10\%$ | $C = 100\%$ | $C = 10\%$ | $C = 100\%$ | $C = 10\%$ | $C = 100\%$ | $C = 100\%$ | $C = 100\%$ |
| FedAvg | 54.78 | 56.82 | 44.11 | 54.37 | 25.77 | 26.73 | 34.47 | 39.93 | 97.54 | 98.59 |
| FedAlt | 56.41 | 56.77 | 69.50 | 64.80 | 15.19 | 10.56 | 28.30 | 26.53 | 97.37 | 99.21 |
| FedSim | 59.90 | 56.07 | 63.46 | 38.34 | 14.80 | 10.46 | 27.00 | 26.47 | 98.63 | 99.60 |
| FedBaBU | 53.12 | 54.60 | 39.77 | 53.21 | 16.77 | 17.33 | 25.60 | 32.47 | 98.19 | 99.07 |
| Ditto | 46.86 | **79.60** | 31.65 | 60.75 | 27.16 | **42.93** | 25.38 | 35.27 | 98.03 | 95.51 |
| FedRep | 53.38 | 57.97 | 47.77 | 67.32 | 34.67 | 31.04 | 24.95 | 39.96 | 97.04 | 98.71 |
| Per-FedAvg | 39.37 | 45.03 | 10.00 | 48.13 | 32.67 | 39.01 | 8.71 | 41.21 | 98.32 | 50.34 |
| Lg-FedAvg | 62.28 | 62.99 | 62.46 | 71.73 | 28.75 | 28.03 | 33.75 | 45.76 | 97.65 | 98.82 |
| **pMixFed** | **65.30** | 75.49 | **74.36** | **75.06** | **34.66** | 41.56 | **43.47** | **51.46** | **99.88** | **99.98** |

function designed for each client. Our results show that the average test accuracy of *pMixFed* outperforms baseline methods in most cases. As shown in Figure 3, the accuracy curve of *pMixFed* exhibits smoother and faster convergence, which may suggest the potential for early stopping in FL settings. Previous studies on mixup also suggest that linear interpolation between features provides the most benefit in the early training phase Zou et al. (2023). Moreover, Figure 3 highlights that partial models with a hard split are highly sensitive to hyperparameter selection and different distribution settings, which can lead to training instability. This issue appears to be addressed more effectively in *pMixFed*, as discussed further in Section 5.3. According to the results in Table 2 and Table 1, *Ditto* demonstrates relatively robust performance across different heterogeneity settings. However, its effectiveness diminishes as the model depth increases, such as with MobileNet, and under larger client populations ($N = 100$). On the other hand, while *FedAlt* and *FedSim* report above-baseline results, they consistently fail during training. [5]

---

[5] Adjusting hyperparameters did not resolve this issue.

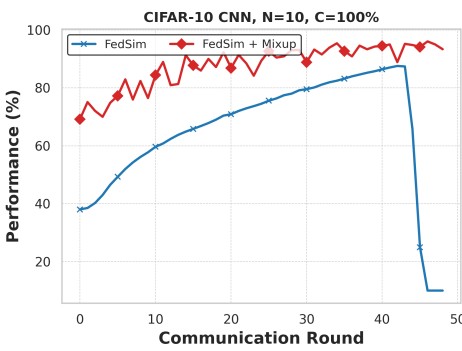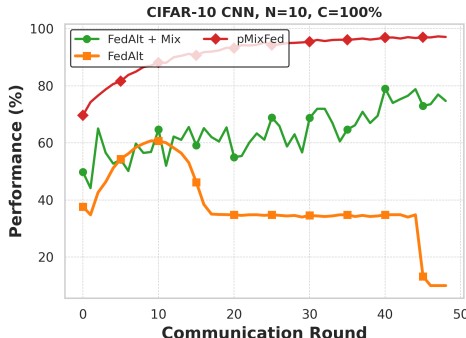

Figure 4: **(a)** The accuracy drop in FedSim occurred due to the vanishing gradient at round 42. **(b)** accuracy declines at round 10 in FedAlt due to the introduction of 5 new participants. Applying adaptive mixup solely between corresponding global and local shared layers mitigates the accuracy drop.

**Caltech-101.** To further evaluate robustness under high inter-class variability and limited per-class samples, we conduct experiments on Caltech-101, a canonical benchmark for heterogeneous federated learning. This dataset poses unique challenges due to its long-tail distribution (most categories have ∼50 images) and high intra-class variability. We follow the same federated setup as in CIFAR/MNIST experiments, splitting the data among $N = \{10, 100\}$ clients with participation rates $C = \{10\%, 100\%\}$. As in prior work, each client is restricted to at most $S = 30$ classes. We compare pMixFed against two recent state-of-the-art personalization baselines: **FedRep** Yang et al. (2023), which learns shared representations with personalized heads, and **Per-FedAvg** Fallah et al. (2020), which meta-learns a global initialization for fast personalization. We also include **FedAlt** Pillutla et al. (2022) as a representative partial-sharing baseline.

Table 3: **Caltech-101** top-1 accuracy (%) with a CNN backbone across four federated settings. Values are the average across evaluation angles A–D; full A–D breakdowns are reported in Appendix E.3.

| Method | $N$=100, $C$=100% | $N$=10, $C$=100% | $N$=100, $C$=10% | $N$=10, $C$=10% |
|---|---|---|---|---|
| **pMixFed (Ours)** | **79.3** | **82.4** | **76.1** | **79.0** |
| FedRep | 77.0 | 80.1 | 74.2 | 76.9 |
| Per-FedAvg | 76.5 | 81.0 | 73.5 | 76.2 |
| FedAlt | 73.1 | 75.5 | 69.5 | 72.2 |

Across all settings, pMixFed consistently achieves the highest accuracy (Table 3). In particular, under the most heterogeneous regime ($N$=100, $C$=10%), pMixFed improves by ∼2% over FedRep and ∼3% over Per-FedAvg, while FedAlt lags by 6–7%. Per-FedAvg performs competitively in the low-client regime ($N = 10$), but its advantage diminishes as heterogeneity increases. These results validate that pMixFed not only excels on CIFAR and MNIST benchmarks, but also generalizes robustly to harder, imbalanced datasets like Caltech-101.

### 5.3   Analytic Experiments

**Adaptive Robustness to Performance Degradation:** During our experiments, we observed the algorithm's ability to adapt and recover from performance degradation, especially in challenging scenarios such as gradient vanishing, adding new users, or introducing unseen incoming data. For instance, in complex settings with larger models, such as *MobileNet* on the CIFAR-100 dataset, partial PFL models like *FedSim* and *FedAlt* experience sudden accuracy drops due to zero gradients or the incorporation of new participants into the cohort. We believe the reason behind this phenomenon is due to the: 1-local and global model update discrepancy in partial models with strict cut in the middle depicted in Figure 1. This degradation is mitigated by the adaptive mixup coefficient, which dynamically adjusts the degree of personalization based on the local model's performance during both the broadcasting and aggregation stages. Specifically, if the global model $G^{(t)}$ lacks sufficient strength, the mixup coefficient $\mu^{(t)}$ is reduced, decreasing the influence of global model. 2- Catastrophic forgetting which is addressed in *pMixFed* by keeping the historical models

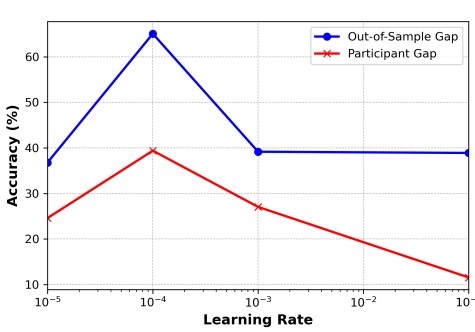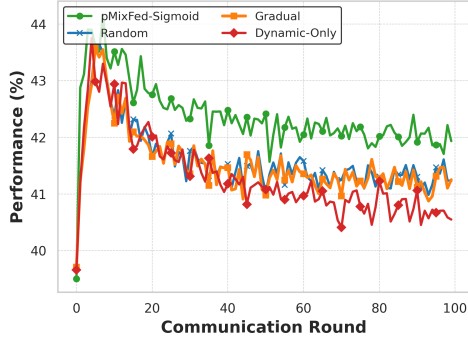

Figure 5: **(a)** Effect on learning rate on average test accuracy(out-of-sample) gap and on the cold-start users. **(b)** The comparison between test accuracy on the cold-start-users with different *Mix factor* functions. **(Dynamic-only)**. In this scenario we used a fixed $Mu$ for all communication rounds. **(Sigmoid)**. The original updating strategy based on a sigmoid function. **(Gradual)**. A simple linear function has been adapted for updating $Mu$. **(Random)** Mixup degree $\lambda_i$ is selected randomly from $\beta$ distribution.

$H\big|_1^T G^t$ in the aggregation process as discussed in section 4.2.2. Figure 4 illustrates that even applying mixup only to the shared layers of the same partial PFL models (FedAlt and Fedsim) enhances resilience against sudden accuracy drops, maintaining model performance over time. In both experiments, the mixup degree for personalized layers is set to $\lambda_i = 0$ for all clients similar to *FedSim and FedAlt* algorithms.

### 5.4   Ablation Study

**Random vs gradual mix factor from $\beta$ distribution:** In this paper, we have explored different designs for calculating g mix factor $\mu_k$. The value of $\lambda$ in Eq. 4, naturally sampled from a $\beta \neq (\alpha, \alpha)$ distribution Zhang et al. (2017) which is on the interval [0,1]. We have also experimented the random $\lambda_i$ using $\beta$ distribution with different $\alpha$. If $\alpha = 1$, the $\beta$ distribution is uniform meaning that the $\lambda$ would be sampled uniformly from [0,1]. Moreover $\alpha > 1$, The $\lambda$ would be more in between, creating a more mixed output between $L_k$ and $G$. On the contrary, if $\alpha < 1$ the mixed model tend to choose just one of the global and local models where $\lambda = 1 or \lambda = 0$. The effects of different $\alpha$ on mixup degree $\lambda$ is discussed in Appendix sec. 4.3

**Mix Factor($\mu$): sigmoid vs Dynamic-only** In this study, we have exploited two different functions to update adaptive mixup factor($\mu$) in each communication round. This idea is based on the performance of the model which we want to update. In first scenario $\sigma$ function has been adapted as shown in equation 5 for adaptively updating mixup degree $\lambda_i$ using sigmoid function. On the other hand, the second scenario, Dynamic-only, parameter $\mu$ is fixed over all communication rounds. The comparison of these two scenarios as well as the effect of different $t$ values on the test accuracy, is depicted in Figure 5 **(b)**.

**Effect of Mixup Degree as Learning Rate (lr):**  We observed that the effect of the mixup coefficient is highly influenced by the learning rate and its decay. To empirically demonstrate this relationship, we measured the impact of the learning rate on new participants (cold-start users) as well as on the out-of-sample gap (average test accuracy on unseen data). The results of this comparison are presented in Figure 5 **(a)**. Additional details are provided in the Appendix sec. 3.4.1. Analytical analysis of the Effect of learning rate and mixup degree.

## 6   Conclusions

We introduced pMixFed, a dynamic, layer-wise personalized federated learning approach that uses mixup to integrate the shared global and personalized local models. Our approach features adaptive partition-

ing between shared and personalized layers, along with a gradual transition for personalization, enabling seamless adaptation for local clients, improved generalization across clients, and reduced risk of catastrophic forgetting. It should be noted that while Mixup has previously been used in FL (e.g., XORMixup, FED-MIX, FedMix), its role was limited to data augmentation. Similarly, model partitioning has been explored in works like FedAlt and FedSim, but only with fixed cut-off layers. In contrast, pMixFed introduces a novel, adaptive integration of layer-wise Mixup directly within the model parameter space—not the feature space. This integration is dynamically updated every communication round based on each client's local accuracy (personalization) relative to the global accuracy (generalization). Governed by a tunable parameter $\mu$, pMixFed enables gradual, client-specific personalization across layers, mitigating global-local discrepancies and client drift in a fully dynamic manner. These innovations distinguish pMixFed from prior approaches, which rely on static strategies. We provided a theoretical analysis of pMixFed to study the properties of its convergence. Our experiments on three datasets demonstrated its superior performance over existing PFL methods. Empirically, pMixFed exhibited faster training times, increased robustness, and better handling of data heterogeneity compared to state-of-the-art PFL models. Future research directions include exploring multi-Modal personalization, exploiting other mixup variants within the parameter space and adapting pMixFed as a dynamic scheduling technique in resource-limited distributed setting.

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

# Appendix

## A  Literature Review

### A.1  Data-Centric

Data-centric methods in PFL seek to mitigate distribution shift and class imbalance by shaping the data seen during training. for example, by adjusting sample sizes, label distributions, and client selection strategies Tan et al. (2022). Common techniques include data normalization, feature engineering, augmentation, synthetic data generation, and adaptive client sampling. Representative examples include Astraea Duan et al. (2019), which performs reputation-based client selection by evaluating neighbors' updates and rewarding trustworthy contributors; P2P k-SMOTE Wang et al. (2021), where clients synthesize minority-class examples via $k$-nearest-neighbor interpolation to reduce local imbalance; and FedMCCS Abdulrahman et al. (2021), which improves fairness by selecting participants based on multiple factors (e.g., dataset size, class diversity, and training loss) rather than at random. FedAug Jeong et al. (2018); Zhao et al. (2018) employs server-side data augmentation to alleviate heterogeneity, while FedHome Wu et al. (2020) augments local data using autoencoders trained with server-provided synthetic samples to address scarcity and imbalance. More recent works directly target class imbalance in PFL: Lee et al. Lee & Choi (2024) construct a separate global model per class and let clients personalize by composing these class-wise globals according to local distributions; Liang et al. Liang et al. (2024) (FedReMa) selectively aggregates updates from the most relevant clients to enhance personalization under skewed data; and FedGA Xiao et al. (2024) aligns client gradients before aggregation to reduce imbalance-induced bias and improve minority-class performance.

Despite their promise, data-centric approaches alter the underlying statistics of federated data and may inadvertently introduce bias or erase rare but important patterns. Several methods also rely on proxy or auxiliary server-side data, raising privacy and distribution-mismatch concerns. Finally, the added computation (and sometimes communication) can be burdensome for resource-constrained devices, limiting the practicality of these strategies in real-world PFL deployments.

### A.2  Clustering

Clustering-based methods in PFL address the core challenge of heterogeneity: each client may hold data from markedly different distributions (e.g., distinct users, sensors, or hospitals). Naïvely averaging updates (e.g., FedAvg) can induce negative transfer when distributions diverge. Instead of a single global model, these approaches exploit structure across clients so that those with similar distributions reinforce one another while avoiding distortion from dissimilar updates. Clients thus retain the benefits of collaboration while mitigating negative transfer; in large populations, clustering/graph constructions also reduce complexity relative to fully personalized models. Clustering methods can be grouped into three categories: (1) *Dynamic Clustering and Graph-based Aggregation*; (2) *Per-Client Weighting*; and (3) *Personalized Clustering.*

In the first category, methods dynamically group clients or build graphs to personalize aggregation based on inter-client relationships. For example, FEDCEDAR Wang et al. (2024) builds dynamic weighted graphs among clients, enabling precise, personalized model distribution via dynamic graph propagation. Multi-Center FL Long et al. (2023) learns multiple global model "centers," assigning each client to its best-fitting center through an EM-style procedure, thereby improving personalization under heterogeneous distributions. More recently, Song et al. (2025) proposes a three-layer clustered hierarchical PFL framework that adapts clustering at multiple levels to handle severe non-IID data.

In the second category (Per-Client Weighting), instead of uniform averaging, methods assign personalized aggregation weights according to client similarity or utility. FedSPD Lin et al. (2024) uses soft clustering so that clients converge toward cluster-relevant models while preserving personalization through flexible model consensus. FedDWA Liu et al. (2023) employs a parameter server to compute dynamic, per-client aggregation weights from model-update similarity, reducing communication and privacy overhead.

In the third category (Personalized Clustering), personalization is organized around classifier outputs or by merging multiple personalized models. For instance, pFedCk Zhang & Shi (2024) combines clustering with knowledge distillation: each client keeps a personalized model locally while an interaction model is shared on the server; features and logits are distilled across similar clients (clustered by the server) to enhance

personalization. Likewise, COMET Cho et al. (2023) clusters clients based on distribution similarity and enables knowledge transfer through logits-based co-distillation, yielding cluster-specific personalized models while maintaining model heterogeneity and communication efficiency.

Clustering can be computationally heavy, and assignments may be unstable—clients can be misassigned when data are noisy or highly imbalanced. Maintaining multiple cluster models becomes costly at scale. Finally, similarity measures or EM responsibilities may leak distribution information, especially in centroid-based or hierarchical clustering.

## A.3   Global Model Adaptation:

In these approaches, a single global model is maintained on the server and subsequently adapted to individual local models in a later phase. The primary goals are: (1) learning a robust, generalized model and (2) enabling fast, efficient local adaptation. Several techniques employ regularization terms to mitigate client drift and prevent model divergence during local updates. For example, FedProx Li et al. (2020) adds a proximal term so local updates stay close to the current global; SCAFFOLD Karimireddy et al. (2020) uses control variates to cancel client drift while training a single server model; MOON Li et al. (2021a) introduces a contrastive term to keep local representations aligned with the global and/or the previous local model; and FedCurv Shoham et al. (2019) infuses an EWC-style curvature penalty into the objective to keep local steps near a shared optimum.

Another line of single-global PFL approaches leverages meta-learning and transfer learning. Methods such as PerFedAvg Fallah et al. (2020) learn a global initialization that each client can adapt with a few steps (MAML in FL). In FedL2P Lee et al. (2023a) meta-nets learn client-specific personalization hyperparameters (e.g., layerwise LRs / BN) to fine-tune the same global efficiently. MetaVers Lim et al. (2024) meta-learns a versatile representation so the shared model adapts rapidly at clients. FedSteg Yang et al. (2020) uses federated transfer/knowledge-transfer ideas for image steganalysis, and DPFed Yu et al. (2020) explicitly studies fine-tuning/local adaptation of a federated global model.

Global model adaptation introduces specific challenges. Meta-learning methods can be computationally intensive, while regularization-based techniques add overhead by incorporating extra terms into the objective. Similarly, transfer-learning approaches can be communication-inefficient and often require a public dataset to enhance the server-side global model. A common limitation across most adaptation techniques is the need for a uniform model architecture across all clients, forcing devices with varying computational capabilities to use the same model size.

Another relevant direction is zero-shot learning (ZSL), which aims to train a model that generalizes on unseen classes, tasks, or domains without having direct training data for them Hao et al. (2021); Zhang et al. (2022); Rostami et al. (2022); Asif et al. (2024); Guo et al. (2025). In the context of federated learning, ZSL can be interpreted as enabling the global or personalized models to perform well for new clients or unseen data distributions without requiring additional local training rounds. Recent works have leveraged generative models Asif et al. (2024) to bridge the gap between seen and unseen tasks. Integrating ZSL capability into PFL could improve support for cold-start clients by allowing them to benefit from global knowledge without significant local updates.

## A.4   Local Model Personalization:

The limitations of existing PFL methods have led to approaches that train customized models for each client. One line of work utilizes multi-task learning (MTL), a collaborative framework that facilitates information exchange across distinct tasks. for example, MOCHA Smith et al. (2017), a canonical federated MTL method that trains related but distinct client models with a shared regularizer; and FedAMP Huang et al. (2021), where attentive message passing builds client-specific models with pairwise collaboration. In Ditto Li et al. (2021b), each client optimizes its own personalized model with a regularization term. FEDHCA2 Lu et al. (2024b) learns relationships between heterogeneous clients to produce personalized but collaborative models. In FedRes Agarwal et al. (2020), clients learn local residuals that add to a shared global model; the personalized local predictor (global + local) can be viewed through the lens of MTL or parameter decoupling which is fully discussed below. Another category leverages knowledge distillation (KD) to support personalization when client-specific training objectives differ. In FedMD Li & Wang (2019),

clients keep heterogeneous local models and distill via public logits to learn personalized local models. In FedGen Zhu et al. (2021), a server-side generator helps clients personalize via KD and feature transfer. FedGKT He et al. (2020) provides group teacher–student training and supports client-side students to form personalized models. These KD-based methods often include an adaptive fusion step that balances generalization and personalization without requiring full model sharing. Some approaches also combine KD with other categories; for example, pFedCK Zhang & Shi (2024) combines client clustering and KD to produce personalized client models, and FedGMKD Zhang et al. (2024) uses prototype-based KD with client-specific distillation to target local personalization.

A third, fast-growing line is hypernetwork or weight-generation–based personalization. Although the number of works is still relatively small, the space is expanding with adapters, prompts, and personalized parameter generation. The first paper, pFedHN Shamsian et al. (2021), trains a server-side hypernetwork that generates client-specific parameters from client embeddings; each client then receives its own weights, and personalization requires minimal client training. HyperFLoRA Lu et al. (2024a) trains a hypernetwork that generates LoRA adapter modules personalized to each client for LLM personalization: clients send lightweight statistics, and the server generates per-client adapters.

Both MTL- and KD-based approaches can incur significant computational and communication overhead, limiting scalability for large-scale FL deployments on resource-constrained devices. Many KD-based PFL methods assume access to proxy/public datasets for logit matching or generator training, which is unrealistic in privacy-sensitive settings. Hypernetworks, on the other hand, face a capacity bottleneck: a single hypernetwork must generate useful weights for many diverse clients; under high heterogeneity, it may underfit or collapse toward "average" solutions.

## B   More Detailed discussion on the Mix Factor

This section elaborates on two key properties of *pMixFed*: 1) dynamic behavior, achieved through a gradual transition in the *mix degree* between layers ($\lambda_i$), and 2) Adaptive behavior, introduced via the *mix factor* ($\mu$). Below, we delve into the details and formulation of each step.

### B.1   Dynamic Mixup Degree

Among the various partitioning strategies for partial PFL discussed in the main paper and in Pillutla et al. (2022), one of the most widely adopted techniques is assigning higher layers of local model $L_{l,k}^t$ to personalization while allowing the base layers $L_{g,k}^t$ to be shared across clients as the global model Oh et al. (2021); Sun et al. (2023); Arivazhagan et al. (2019). This design aligns with insights from the Model-Agnostic Meta-Learning (MAML) algorithm Collins et al. (2022), which demonstrates that lower layers generally retain task-agnostic, generalized features, while the higher layers capture task-specific, personalized characteristics. Accordingly, in this work, we designate the head of the model as containing personalized information, while the base layers represent generalized information shared across clients [6].

### B.1.1   Broadcasting:

To achieve a nuanced and gradual transition in the mixing process between the global model and the local model, we define the mix degree $\lambda_i$ as follows. The local model's head $L_n$ remains frozen, and the head of the global model is excluded from being shared with the local model. The mix degree for each layer $\lambda_i$ increases incrementally based on the *mix factor* $\mu$, such that as we move toward the base layers, the personalization impact decreases. This dynamic behavior is represented as:

$$\lambda_i = \lambda_{i+1} + \mu,$$

where $\lambda_i$ controls the degree of mixing at layer $i$. This process is visually illustrated in Figure 6.

---

[6]It should be noted that our method is fully adaptable to different partial PFL designs as well discussed in Pillutla et al. (2022).

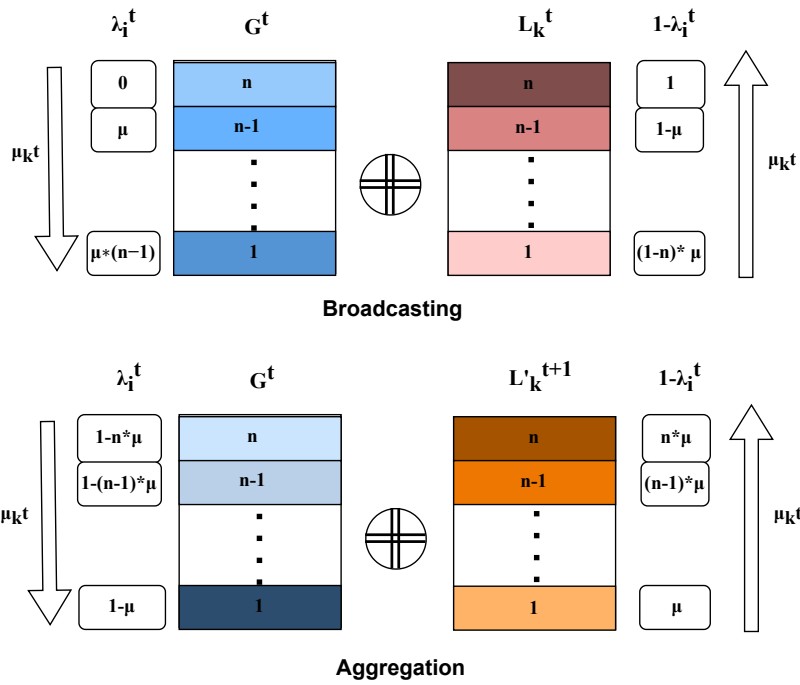

Figure 6: layer-wise Dynamic transition of mixup degree $\lambda_i$ in Broadcasting and aggregation phase. The darker color shows the higher mixup degree ($\lambda_i$) for the corresponding layer $i$.

### B.1.2 Aggregation:

The aggregation stage focuses on preserving the generalized information from the history of previous global models to mitigate the catastrophic forgetting problem. In contrast to the broadcasting process, the primary goal here is to retain the generalized information of the previous global model, which encapsulates the history of all prior models $H\big|_1^{t-1}G$. Therefore, the base layers should predominantly be shared from the global model, particularly during the first round of training, where the updated local models are still underdeveloped. Nonetheless, as we transition to the head, it's less prominent to transfer knowledge from the previous global models. Figure 6 illustrates how the mixup degree transitions from the head to the base layers.

### B.2 Adaptive Mix-Factor

To enable online, adaptive updates of the mix degree $\lambda_i$, we implemented several algorithms, as detailed in Section *5.4 Ablation Study*. We observed a clear relationship between the mixup degree and the similar behavior to lr, which is further discussed in Section **??**. Inspired by learning rate schedulers, we introduced the mix-factor $\mu$ to adaptively update the layer-wise mixup degree based on the current communication round $\delta = t/T$ and the relative performance of the current local model $L_k^t$ compared to the global model $G^t$. The Sigmoid function in Figure 6 illustrates how $\mu$ evolves with respect to $\delta$ and accuracy ($Acc$). The best results were achieved when setting $b = 1$ and using the square of $Acc$ as an exponent. The rationale behind this approach is that a more experienced, better-performing model should share more information. Specifically, if the local model accuracy $Acc_l$ significantly exceeds that of the global model ($Acc_G$) in the current round ($t$) such that $Acc \gg 1$, less information is shared from the global model. Conversely, when $0 < Acc < 1$, the global model dominates the parameter updates in both the global and local models. Figure 7 shows the distribution of the calculated $\mu$ across different communication rounds for client 0 in the broadcasting stage. In the broadcasting stage, higher $Acc_l$ values result in freezing more layers for personalization, leading to a decrease in $\mu$. Conversely, during the aggregation phase, if the global model accuracy $Acc_G$ outperforms

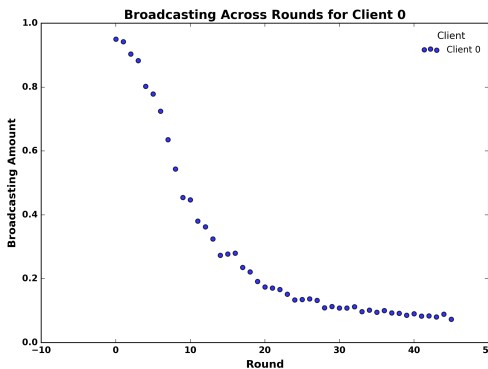

Figure 7: mix-factors for client 0 in different communication rounds

the updated local model accuracy $Acc_{l'}^{(t+1)}$, more base layers are shared. This is especially important during the first round of training, where local models are less stable. As demonstrated in *Section 5.3, our proposed method adapts dynamically to performance drops and high-distribution complexity, adjusting the mixup degree as needed. This adaptability is effective even when applied partially to partial PFL methods such as FedAlt, and FedSim.*

### B.3 Model Heterogeneity

*pMixFed* is capable of handling variable model sizes across different clients. The global model, $M_g$, retains the maximum number of layers from all clients, i.e., $M_G = \max(M_1, M_2, \ldots, M_N)$. During the matching process between the global model $G_i$ and the local model $L_i$, if a layer block from the local model does not match a corresponding global layer, we set $\lambda_i = 0$, meaning that the layer block will neither participate in the broadcasting nor aggregation processes. So the layers are participating according to their existing participation rate. For instance, if only 40% of clients have more than 4 layers, the generalization degree (in both broadcasting and aggregation stages), will be less than 0.4 in each training round due to different participation rates.

### B.4 Difference from other works:

While Mixup has previously been used in FL (e.g., XORMixup, FEDMIX, FedMix), its role was limited to data augmentation. Similarly, model partitioning has been explored in works like FedAlt and FedSim, but only with fixed cut-off layers. In contrast, pMixFed introduces a novel, adaptive integration of layer-wise Mixup directly within the model parameter space—not the feature space. This integration is dynamically updated every communication round based on each client's local accuracy (personalization) relative to the global accuracy (generalization). Governed by a tunable parameter $\mu$, pMixFed enables gradual, client-specific personalization across layers, mitigating global-local discrepancies and client drift in a fully dynamic manner. These innovations distinguish pMixFed from prior approaches, which rely on static strategies.

## C Theoretical Analysis

In this section we establish a clean theoretical view of *pMixFed*. We show that the update rule approximates SGD dynamics on the global objective with an *effective step size* governed by the mixup coefficients. This both clarifies stability conditions and explains how pMixFed mitigates catastrophic forgetting. In *FedSGD*, the gradients are aggregated and the server will be update the global model according to the aggregated gradients. *FedSGD* is sometimes preferred over *FedAvg* due to its potentially faster convergence. However, it lacks robustness in heterogeneous environments. *pMixFed* leverages the faster convergence characteristics of *FedSGD* by incorporating early stopping mechanisms, facilitated by the use of mixup. As It will be discussed in Section 5.4, the mixup factor $\lambda$ functions analogously to an SGD update at the server, even

Table 4: Notation used throughout the paper. Vectors are column vectors; $\|\cdot\|$ is the Euclidean norm.

| Symbol | Type | Meaning |
|---|---|---|
| $K$ | scalar | Number of clients |
| $\mathcal{U}_t$ | set | Clients selected at round $t$; $|\mathcal{U}_t|=K$ |
| $D_k$ | dataset | Local dataset of client $k$; size $|D_k|$ |
| $|D|$ | scalar | Total data size, $|D|=\sum_k |D_k|$ |
| $\Omega_k$ | scalar | Aggregation weight $|D_k|/|D|$ |
| $\ell(\cdot,\cdot)$ | function | Pointwise loss used to build local/global objectives |
| $f_k(\theta,x)$ | function | Model prediction on client $k$ with parameters $\theta$ |
| $F_k(\theta)$ | function | Local objective on client $k$ |
| $F(\theta)$ | function | Global objective $\sum_k \Omega_k F_k(\theta)$ |
| $\theta$ | vector | Global (server) model parameters |
| $\theta_k$ | vector | Client $k$'s local/personal parameters (if stored) |
| $G^{(t)}$ | vector | Server/global parameters at round $t$ (alias of $\theta^t$) |
| $L_k^{(t)}$ | vector | Local model of client $k$ at round $t$ |
| $L_{g,k}^{(t)}$ | vector | Global/shared layers of $L_k^{(t)}$ |
| $L_{l,k}^{(t)}$ | vector | Local/personalized layers of $L_k^{(t)}$ |
| $M$ | scalar | Number of layers in the network |
| $s$ | index | Split (cut) layer index, $1 \le s \le M$ |
| $t$ | index | Communication round index ($t=0,\ldots,T-1$) |
| $T$ | scalar | Total number of communication rounds |
| $r,\ \tau$ | scalar | Number of local steps per round (local iterations/epochs) |
| $b$ | scalar | Local mini-batch size |
| $\eta_\ell$ | scalar | Local learning rate (client step size) |
| $\eta_g$ | scalar | Server/aggregation step size in the FedSGD view |
| $\lambda_{k,i}^t$ | scalar | Mixup coefficient for client $k$, layer $i$, round $t$ ($\in [0,1]$) |
| $\lambda_k^t$ | scalar | Client-level mixup (uniform across layers if assumed) |
| $\bar{\lambda}^t$ | scalar | Population-averaged mixup, $\sum_{k\in\mathcal{U}_t} \Omega_k \lambda_k^t$ |
| $\eta_{\text{eff}}^t$ | scalar | Effective step size $(1-\bar{\lambda}^t)\eta_\ell$ in SGD view |
| $\mu$ | scalar | Mix factor controlling the schedule of $\lambda$ across layers/rounds |
| $\nabla F_k(\theta)$ | vector | Full local gradient on client $k$ |
| $g_k^t$ | vector | Stochastic gradient on client $k$ at round $t$ |
| $g_{k,s}^t$ | vector | Stochastic gradient at local step $s\in\{0,\ldots,\tau-1\}$ in round $t$ |
| $L$ | scalar | Smoothness constant (Lipschitz gradient) |
| $\mu_{\text{sc}}$ | scalar | Strong convexity constant (when assumed in Sec. 4) |
| $\kappa$ | scalar | Condition number $L/\mu_{\text{sc}}$ (strongly convex case) |
| $\sigma^2$ | scalar | Upper bound on gradient variance |
| $\zeta_k$ | scalar | Gradient dissimilarity: $\sup_\theta \|\nabla F(\theta)-\nabla F_k(\theta)\|^2$ |
| $\zeta$ | scalar | Aggregate dissimilarity (e.g., $\sum_k \zeta_k$ or mean variant) |
| $\Delta_k$ | scalar | Local–global optimum gap $\|\theta_k^\star-\theta^\star\|^2$ |
| $\mathcal{G}$ | scalar | Gradient moment bound used in multi-step analysis (Sec. 4) |
| $\text{Beta}(\alpha_{\text{B}},\alpha_{\text{B}})$ | dist. | Beta distribution used to sample $\lambda$ in MixUp |

**Notes.** (i) Aggregation-as-FedSGD yields $\boxed{\bar{\lambda}^t = 1 - \eta_g/\eta_\ell}$ and $\eta_{\text{eff}}^t = (1-\bar{\lambda}^t)\eta_\ell$. (ii) We reserve $\mu_{\text{sc}}$ for strong convexity; $\mu$ denotes the mixup scheduling factor if present in Methodology.

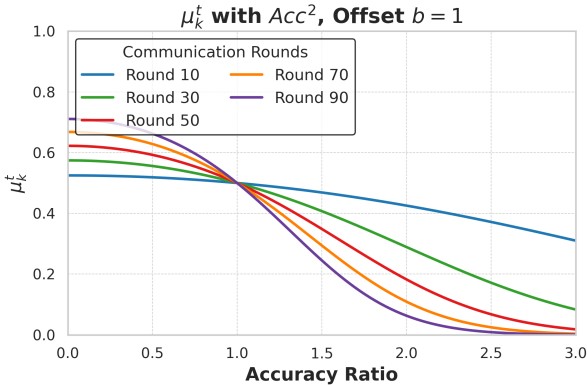

Figure 8: Adaptive mix-factor $\mu_k^t$ according to the accuracy ration $Acc^t = Acc_L^t/Acc_G^t$ in different communication rounds.

though *pMixFed* aggregates model weights rather than gradients, similar to *FedAvg*. Section C.2 provides a more detailed explanation of this mechanism.

## C.1 Assumptions

Let $F(\theta) := \sum_{k=1}^{K} \Omega_k F_k(\theta)$ denote the global objective, where $\Omega_k = |D_k|/|D|$ and $\theta^t$ is the global model at round $t$. Each client $k$ runs $r$ local SGD steps with step size $\eta_\ell$, producing $\theta_k^{t+1}$. Mixup coefficients $\lambda_{k,i}^t \in [0,1]$ apply at layer $i$ of client $k$. For simplicity we define client-averaged $\lambda_k^t := \sum_i w_i \lambda_{k,i}^t$ with $\sum_i w_i = 1$, and population average $\bar{\lambda}^t := \sum_k \Omega_k \lambda_k^t$. We make the following standard assumptions. These assumptions are standard in convergence analysis and ensure that the optimization process is well-behaved.

**Assumption 1.** *(Unbiased Gradient Estimation with Bounded Variance.) The gradient estimate for local model updates is unbiased and has a bounded variance, i.e.,)*

$$\mathbb{E}[g_k^t] = \nabla F_k(\theta^t) \quad and \quad \mathbb{E}\|g_k^t - \nabla F_k(\theta^t)\|^2 \leq \sigma^2. \tag{9}$$

**Assumption 2** (L-Smoothness). *Each $F_k$ has L-Lipschitz gradients, i.e.,*

$$\|\nabla F_k(x) - \nabla F_k(y)\| \leq L\|x - y\| \tag{10}$$

**Assumption 3** (Bounded Gradients). *The gradients at each client are bounded, i.e.,*

$$\mathbb{E}\|g_k^t\|^2 \leq G^2 \tag{11}$$

## C.2 Similarities to FedSGD

In *pMixFed*, the server forms a convex combination between the previous global iterate and the locally updated models:

$$\theta^{t+1} = \sum_{k\in\mathcal{U}_t} \Omega_k \left(\lambda_k^t \theta^t + (1-\lambda_k^t)\theta_k^{t+1}\right) = \bar{\lambda}^t \theta^t + (1-\bar{\lambda}^t)\sum_{k\in\mathcal{U}_t}\Omega_k\theta_k^{t+1}. \tag{12}$$

With one local step from $\theta^t$, $\theta_k^{t+1} = \theta^t - \eta_\ell g_k^t$, (12) becomes

$$\theta^{t+1} = \theta^t - \underbrace{(1-\bar{\lambda}^t)\eta_\ell}_{\eta_{\text{eff}}^t} \sum_{k\in\mathcal{U}_t} \Omega_k g_k^t. \tag{13}$$

Thus *pMixFed* is acting very similar to FedSGD *F* with *effective* step size

$$\boxed{\eta_{\text{eff}}^t = (1-\bar{\lambda}^t)\eta_\ell.}$$

**Matching the FedSGD view.** If the server instead writes a FedSGD step with global step size $\eta_g > 0$, i.e., $\theta^{t+1} = \theta^t - \eta_g \sum_k \Omega_k g_k^t$, then (13) shows the coefficients must satisfy:

**Theorem 1** (Coefficient matching with FedSGD). *For any round $t$, the pMixFed update (12) similar with a FedSGD step of size $\eta_g$ if and only if*

$$\bar{\lambda}^t = 1 - \frac{\eta_g}{\eta_\ell}, \qquad 0 \leq \frac{\eta_g}{\eta_\ell} \leq 1.$$

*Proof.* Equate (13) with $\theta^{t+1} = \theta^t - \eta_g \sum_k \Omega_k g_k^t$ and match the coefficient of the aggregated gradient: $(1 - \bar{\lambda}^t)\eta_\ell = \eta_g$. $\square$

**Interpretation.** Larger $\eta_g/\eta_\ell$ implies smaller $\bar{\lambda}^t$, i.e., less carry-over of $\theta^t$ and more trust in the new local updates. Standard smoothness suggests choosing $\eta_{\text{eff}}^t \leq 1/L$, equivalently $\bar{\lambda}^t \geq 1 - \frac{1}{L\eta_\ell}$ whenever $L\eta_\ell > 1$.

## C.3 Convergence guarantees

The SGD view (13) directly yields standard optimization guarantees. We first state a one-step descent inequality, then derive the nonconvex and strongly-convex global results.

**Lemma 1** (One-step descent). *Under (Asuumption1)–(Assumption2) and $\eta_{\text{eff}}^t \leq 1/L$,*

$$\mathbb{E}\big[F(\theta^{t+1})\big] \leq \mathbb{E}\big[F(\theta^t)\big] - \frac{\eta_{\text{eff}}^t}{2} \mathbb{E}\big\|\nabla F(\theta^t)\big\|^2 + \frac{L(\eta_{\text{eff}}^t)^2}{2} \sigma^2.$$

*Proof.* By $L$-smoothness, $F(\theta^{t+1}) \leq F(\theta^t) + \langle \nabla F(\theta^t), \theta^{t+1} - \theta^t \rangle + \frac{L}{2}\|\theta^{t+1} - \theta^t\|^2$. Substitute (13), take conditional expectation using $\mathbb{E}[\sum_k \Omega_k g_k^t] = \nabla F(\theta^t)$ and $\mathbb{E}\|\sum_k \Omega_k g_k^t\|^2 \leq \|\nabla F(\theta^t)\|^2 + \sigma^2$, and use $\eta_{\text{eff}}^t \leq 1/L$ to absorb the quadratic term in $\|\nabla F(\theta^t)\|^2$. $\square$

**Theorem 2** (Nonconvex rate). *Let $\eta_{\text{eff}}^t \equiv \eta_{\text{eff}} \leq 1/L$ be constant. Then for any $T \geq 1$,*

$$\frac{1}{T} \sum_{t=0}^{T-1} \mathbb{E}\big\|\nabla F(\theta^t)\big\|^2 \leq \frac{2(F(\theta^0) - F^\star)}{T\,\eta_{\text{eff}}} + L\,\eta_{\text{eff}}\,\sigma^2.$$

*Proof.* Sum Lemma 1 over $t = 0, \ldots, T-1$ and telescope; divide by $T$. Please see appendix . D $\square$

**Theorem 3** (Strongly-convex case). *Under (A1)–(A3), with $\eta_{\text{eff}}^t \equiv \eta_{\text{eff}} \leq \min\{1/L, 1/(2\mu)\}$,*

$$\mathbb{E}\big\|\theta^t - \theta^\star\big\|^2 \leq (1 - \mu\,\eta_{\text{eff}})^t \big\|\theta^0 - \theta^\star\big\|^2 + \mathcal{O}\bigg(\frac{\sigma^2}{\mu}\,\eta_{\text{eff}}\bigg).$$

*Proof.* Apply the standard strongly-convex SGD recursion with step $\eta_{\text{eff}}$ to (13). Please see appendix . D $\square$

## C.4 Layer-wise mixing and forgetting control

Layer-wise coefficients $\lambda_{k,i}^t$ (with averaging weights $w_i$) produce the same effective step via $\bar{\lambda}^t = \sum_{k,i} \Omega_k w_i \lambda_{k,i}^t$ and $\eta_{\text{eff}}^t = (1 - \bar{\lambda}^t)\eta_\ell$. The per-round drift satisfies

$$\big\|\theta^{t+1} - \theta^t\big\| = \eta_{\text{eff}}^t \bigg\|\sum_{k \in \mathcal{U}_t} \Omega_k\,g_k^t\bigg\|.$$

Hence larger $\bar{\lambda}^t$ explicitly caps the amount a single round can overwrite the global representation, limiting catastrophic forgetting, while smaller $\bar{\lambda}^t$ accelerates adaptation. Maintaining $\eta_{\text{eff}}^t \leq 1/L$ ensures stability irrespective of the layer schedule.

### C.5  Multiple local steps per round

Suppose each selected client performs $\tau \geq 1$ local SGD steps from $\theta^t$: $\theta_k^{t+1} = \theta^t - \eta_\ell \sum_{s=0}^{\tau-1} g_{k,s}^t$. Then

$$\theta^{t+1} = \theta^t - (1 - \bar{\lambda}^t)\eta_\ell \sum_{k \in \mathcal{U}_t} \Omega_k \sum_{s=0}^{\tau-1} g_{k,s}^t \equiv \theta^t - \eta_{\text{eff},\tau}^t \widehat{g}^t, \qquad \eta_{\text{eff},\tau}^t := (1 - \bar{\lambda}^t)\eta_\ell \tau,$$

where $\widehat{g}^t$ is the average of the $\tau$ local gradients. Under (Assumption1)–(Assumption2) and bounded second moments, standard local-SGD arguments give

$$\left\| \mathbb{E}[\widehat{g}^t] - \nabla F(\theta^t) \right\| \leq c_1 L \eta_\ell (\tau - 1) G,$$

for a constant $c_1$ and gradient-moment bound $G$. Consequently, *pMixFed* behaves like SGD with step $\eta_{\text{eff},\tau}^t$ up to a bias of order $(1 - \bar{\lambda}^t) L \eta_\ell^2 \tau(\tau - 1)$. Choosing $\bar{\lambda}^t$ so that $\eta_{\text{eff},\tau}^t \leq 1/L$ recovers the guarantees of Theorems 2–3 with an additional (controlled) residual proportional to $L \eta_\ell^2 \tau(\tau - 1)$.

**Remark1.** *Theorem 1 establishes a direct relationship between the mixup coefficient $\lambda_k$ and the learning rates used in local and global updates. This insight allows us to interpret the mixup mechanism in pMixFed as adjusting the effective learning rate at the server, providing a theoretical foundation for selecting $\lambda_k$ based on desired convergence properties.*

**Remark2.** *In practice, this relationship suggests that by tuning $\lambda_k$, we can control the influence of the global model versus the local models in the aggregation process, similar to adjusting the learning rate in FedSGD. This is particularly beneficial in heterogeneous environments where clients may have varying data distributions.*

Our theoretical analysis indicates that the mixup coefficient $\lambda_k$ in *pMixFed* plays a role analogous to the learning rate in *FedSGD*. This equivalence provides a deeper understanding of how *pMixFed* leverages the strengths of *FedSGD* while mitigating its weaknesses in heterogeneous settings. By appropriately choosing $\lambda_k$, *pMixFed* can achieve faster convergence and improved robustness. The *pMixFed* algorithm combines the advantages of FedSGD and FedAvg by aggregating model weights rather than gradients, while still ensuring faster convergence even in heterogeneous data settings due to it's adaptation ability. The incorporation of the mixup mechanism enhances stability, providing faster convergence rates compared to FedSGD, particularly in non-convex settings

## D  Proof of Theorems:

**Proof of Theorem 1 (SGD form with an effective step).**  By definition of the PMIXFED aggregation,

$$\theta^{t+1} = \sum_{k \in \mathcal{U}_t} \Omega_k \left( \lambda_k^t \theta^t + (1 - \lambda_k^t) \theta_k^{t+1} \right) = \bar{\lambda}^t \theta^t + (1 - \bar{\lambda}^t) \sum_{k \in \mathcal{U}_t} \Omega_k \theta_k^{t+1},$$

where $\bar{\lambda}^t := \sum_{k \in \mathcal{U}_t} \Omega_k \lambda_k^t \in [0, 1]$. Each selected client performs one local SGD step from the broadcasted model $\theta^t$: $\theta_k^{t+1} = \theta^t - \eta_\ell g_k^t$, where $g_k^t$ is the stochastic gradient computed on client $k$ at $\theta^t$. Substituting into the aggregation gives

$$\theta^{t+1} = \bar{\lambda}^t \theta^t + (1 - \bar{\lambda}^t) \sum_{k \in \mathcal{U}_t} \Omega_k \left( \theta^t - \eta_\ell g_k^t \right) = \theta^t - (1 - \bar{\lambda}^t)\eta_\ell \sum_{k \in \mathcal{U}_t} \Omega_k g_k^t.$$

Therefore the server update is *exactly* an SGD step on $F(\theta) = \sum_k \Omega_k F_k(\theta)$ with the *effective* step size $\eta_{\text{eff}}^t := (1 - \bar{\lambda}^t)\eta_\ell$, i.e., $\theta^{t+1} = \theta^t - \eta_{\text{eff}}^t \sum_{k \in \mathcal{U}_t} \Omega_k g_k^t$.

**Proof of Theorem 2 (Coefficient matching with FedSGD).** Assume the server writes its update as a FedSGD step with server stepsize $\eta_g > 0$: $\theta^{t+1} = \theta^t - \eta_g \sum_{k \in \mathcal{U}_t} \Omega_k\, g_k^t$. From Theorem 1 we also have the PMIXFED SGD form $\theta^{t+1} = \theta^t - (1 - \bar{\lambda}^t)\, \eta_\ell \sum_{k \in \mathcal{U}_t} \Omega_k\, g_k^t$. Equality of the two updates for arbitrary realizations of the stochastic gradients holds if and only if their coefficients on the aggregated gradient coincide, i.e.,

$$(1 - \bar{\lambda}^t)\, \eta_\ell \;=\; \eta_g \qquad \Longleftrightarrow \qquad \bar{\lambda}^t \;=\; 1 - \frac{\eta_g}{\eta_\ell}.$$

Since by construction $\bar{\lambda}^t \in [0, 1]$, we immediately obtain the admissible range $0 \le \eta_g/\eta_\ell \le 1$. This proves the necessary and sufficient condition for coefficient matching.

**Proof of Theorem 3 (Nonconvex rate).** Recall the PMIXFED update in SGD form

$$\theta^{t+1} \;=\; \theta^t \;-\; \eta_{\mathrm{eff}} \underbrace{\sum_{k \in \mathcal{U}_t} \Omega_k\, g_k^t}_{:=\, \bar{g}^t}, \qquad \eta_{\mathrm{eff}} \;\le\; \frac{1}{L},$$

where by (A2) the aggregated stochastic gradient $\bar{g}^t$ is unbiased, $\mathbb{E}[\bar{g}^t \mid \theta^t] = \nabla F(\theta^t)$, and has bounded second moment $\mathbb{E}\|\bar{g}^t - \nabla F(\theta^t)\|^2 \le \sigma^2$. By $L$-smoothness (A1),

$$F(\theta^{t+1}) \;\le\; F(\theta^t) + \langle \nabla F(\theta^t),\, \theta^{t+1} - \theta^t \rangle + \frac{L}{2}\, \|\theta^{t+1} - \theta^t\|^2.$$

Substitute $\theta^{t+1} - \theta^t = -\eta_{\mathrm{eff}}\, \bar{g}^t$, take conditional expectation given $\theta^t$, and use $\mathbb{E}\|\bar{g}^t\|^2 = \|\nabla F(\theta^t)\|^2 + \mathbb{E}\|\bar{g}^t - \nabla F(\theta^t)\|^2 \le \|\nabla F(\theta^t)\|^2 + \sigma^2$ to obtain

$$\mathbb{E}\big[F(\theta^{t+1}) \,\big|\, \theta^t\big] \;\le\; F(\theta^t) \;-\; \eta_{\mathrm{eff}}\, \|\nabla F(\theta^t)\|^2 \;+\; \frac{L\, \eta_{\mathrm{eff}}^2}{2}\Big(\|\nabla F(\theta^t)\|^2 + \sigma^2\Big).$$

Rearranging,

$$\mathbb{E}\big[F(\theta^{t+1}) \,\big|\, \theta^t\big] \;\le\; F(\theta^t) \;-\; \Big(\eta_{\mathrm{eff}} - \tfrac{L\eta_{\mathrm{eff}}^2}{2}\Big) \|\nabla F(\theta^t)\|^2 \;+\; \frac{L\, \eta_{\mathrm{eff}}^2}{2}\, \sigma^2.$$

Since $\eta_{\mathrm{eff}} \le 1/L$, we have $\eta_{\mathrm{eff}} - \frac{L\eta_{\mathrm{eff}}^2}{2} \ge \frac{\eta_{\mathrm{eff}}}{2}$; taking total expectation yields the one-step descent inequality

$$\mathbb{E}\big[F(\theta^{t+1})\big] \;\le\; \mathbb{E}\big[F(\theta^t)\big] \;-\; \frac{\eta_{\mathrm{eff}}}{2}\, \mathbb{E}\big\|\nabla F(\theta^t)\big\|^2 \;+\; \frac{L\, \eta_{\mathrm{eff}}^2}{2}\, \sigma^2. \qquad (\star)$$

Summing $(\star)$ over $t = 0, \ldots, T-1$ and telescoping gives

$$\frac{\eta_{\mathrm{eff}}}{2} \sum_{t=0}^{T-1} \mathbb{E}\big\|\nabla F(\theta^t)\big\|^2 \;\le\; F(\theta^0) - F^\star \;+\; \frac{L\, \eta_{\mathrm{eff}}^2}{2}\, T\, \sigma^2.$$

Divide both sides by $T\, \eta_{\mathrm{eff}}/2$ to conclude

$$\frac{1}{T} \sum_{t=0}^{T-1} \mathbb{E}\big\|\nabla F(\theta^t)\big\|^2 \;\le\; \frac{2\,(F(\theta^0) - F^\star)}{T\, \eta_{\mathrm{eff}}} \;+\; L\, \eta_{\mathrm{eff}}\, \sigma^2,$$

which is precisely the claimed nonconvex rate.

**Proof of Theorem 4 (Strongly convex case).** Assume (A1)–(A3) and $\eta_{\mathrm{eff}} \le \min\{1/L,\, 1/(2\mu)\}$. Starting again from the smoothness inequality and the SGD update,

$$\mathbb{E}\big[F(\theta^{t+1}) \,\big|\, \theta^t\big] \;\le\; F(\theta^t) \;-\; \eta_{\mathrm{eff}}\, \|\nabla F(\theta^t)\|^2 \;+\; \frac{L\, \eta_{\mathrm{eff}}^2}{2}\Big(\|\nabla F(\theta^t)\|^2 + \sigma^2\Big).$$

As before, with $\eta_{\text{eff}} \leq 1/L$ we get

$$\mathbb{E}\big[F(\theta^{t+1}) \,\big|\, \theta^t\big] \;\leq\; F(\theta^t) \;-\; \frac{\eta_{\text{eff}}}{2}\,\|\nabla F(\theta^t)\|^2 \;+\; \frac{L\,\eta_{\text{eff}}^2}{2}\,\sigma^2. \tag{$\dagger$}$$

Strong convexity implies the Polyak–Łojasiewicz (PL) inequality $\|\nabla F(\theta^t)\|^2 \geq 2\mu\big(F(\theta^t) - F^\star\big)$. Plugging this into ($\dagger$) and taking total expectation gives

$$\mathbb{E}\big[F(\theta^{t+1}) - F^\star\big] \;\leq\; \big(1 - \mu\,\eta_{\text{eff}}\big)\,\mathbb{E}\big[F(\theta^t) - F^\star\big] \;+\; \frac{L\,\eta_{\text{eff}}^2}{2}\,\sigma^2.$$

Unrolling the linear recursion,

$$\mathbb{E}\big[F(\theta^t) - F^\star\big] \;\leq\; (1 - \mu\,\eta_{\text{eff}})^t\big(F(\theta^0) - F^\star\big) \;+\; \frac{L\,\eta_{\text{eff}}\,\sigma^2}{2\mu}.$$

Finally, strong convexity also yields $F(\theta) - F^\star \geq \frac{\mu}{2}\|\theta - \theta^\star\|^2$, so

$$\mathbb{E}\,\|\theta^t - \theta^\star\|^2 \;\leq\; \frac{2}{\mu}\,\mathbb{E}\big[F(\theta^t) - F^\star\big] \;\leq\; (1 - \mu\,\eta_{\text{eff}})^t\,\|\theta^0 - \theta^\star\|^2 \;+\; \mathcal{O}\!\Big(\frac{\sigma^2}{\mu}\,\eta_{\text{eff}}\Big),$$

which is the claimed result.

## E   Additional Details about the Experiments

### E.1   Experimental Setup

For creating heterogenity, we followed the **Training Details:** For evaluation, We have reported the average test accuracy of the global model Yuan et al. (2021) for different approaches. The final global model at the last communication round is saved and used during the evaluation. The global model is then personalized according to each baseline's personalization or fine-tuning algorithm for $r = 4$ local epochs and $T = 50$. For **FedAlt**, the local model is reconstructed from the global model and fine-tuned on the test data. For **FedSim**, both the global and local models are fine-tuned partially but simultaneously. In the case of **FedBABU**, the head (fully connected layers) remains frozen during local training, while the body is updated. Since we could not directly apply **pFedHN** in our platform setting, we adapted their method using the same hyper parameters discussed above and employed hidden layers with 100 units for the hypernetwork and 16 kernels. The local update process for **LG-FedAvg**, **FedAvg**, and **Per-FedAvg** simply involves updating all layers jointly during the fine-tuning process. The global learning rate for the methods that need sgd update in the global server e.g., FedAvg, has been set from $lr_{global} = [1e-3, 1e-4, and 1e-5]$. It should be noted that due to the performance drop for some methods (FedAlt , FedSim) in round 10 or 40 in some settings, we've reported the highest accuracy achieved. Also this is the reason the accuracy curves are illustrated for 39 rounds instead of 50. [7]

### E.2   Caltech-101 Experimental Setup and Results

**Dataset.**   Caltech-101 Fei-Fei et al. (2004) contains 9,146 images across 101 object categories plus a background class, with $\sim$40–800 images per class (most near 50). Images are resized to $224 \times 224$ for CNN training. We followed prior FL literature Fallah et al. (2020); Yang et al. (2023); Pillutla et al. (2022) in simulating heterogeneous splits: each client receives at most $S = 30$ classes sampled by Dirichlet distribution ($\alpha = 0.5$). We evaluate four federated settings: $N = \{10, 100\}$ clients with participation $C = \{10\%, 100\%\}$.

---

[7]As discussed in the main paper, the change of hyper-parameters such as lr, batch size, momentum and even changing the optimizer to adam didn't help with the performance drop in most cases.

**Model and Training.** For consistency with CIFAR/MNIST, we adopt a CNN backbone (4 convolutional layers + 1 fully connected layer). Each client trains for $r = 4$ local epochs per round with batch size 32. The total communication rounds are $T = 50$, using Adam optimizer with learning rate $1 \times 10^{-3}$. For partial baselines (FedAlt), the split layer is fixed at the midpoint. For Per-FedAvg, meta-initialization is updated with inner learning rate 0.01 and outer learning rate 0.001. FedRep uses a shared representation with personalized heads (dimension 256). Evaluation is done by fine-tuning on local test splits for 4 epochs.

**Results.** Table 5 reports detailed top-1 accuracy (%) across four evaluation angles (A–D), along with the average. Across all federated settings, pMixFed achieves the best performance, with particularly strong improvements in the heterogeneous regime ($N = 100$, $C = 10\%$). FedRep performs second-best overall, confirming the utility of representation sharing, while Per-FedAvg is strong when client counts are small. FedAlt trails consistently, highlighting the limitations of fixed partitioning.

### E.3 Results on Caltech-101 dataset

We additionally evaluate on Caltech-101 Fei-Fei et al. (2004), following the same heterogeneous FL setup as CIFAR/MNIST. Each client receives at most $S = 30$ classes, and data is split using a Dirichlet distribution ($\alpha = 0.5$). We report accuracy across four evaluation angles (A–D) and their mean (AVG).

### E.4 Out-of-Sample and Participation Gap

In the evaluation of the effect of learning rate and mixup, the average test accuracy[8] is measured on cold-start clients $|D^{ts}_{k \cap \text{unseen}}|, k \subset \{1, \ldots, M\}$, where $D^{ts} \neq D^{tr}$. These clients have not participated in the federation at any point during training. *FedAlt* and *FedSim* perform poorly on cold-start users or unseen clients, highlighting their limited generalization capability. The test accuracy of *pFedMix*, while affected under a 10% participation rate, benefits significantly from seeing more clients. Increased client participation directly improves accuracy, as observed in previous studies Pillutla et al. (2022).

### E.5 Ablation Study : The Effects of Different alpha on Mixup Degree

The Beta distribution $\beta(\alpha, \alpha)$ is defined on the interval [0,1], where the parameter $\alpha$ controls the shape of the distribution. The value of $\lambda$, used in Eq. 4, is naturally sampled from this distribution. By varying the parameter $\alpha$, we can adjust how much mixing occurs between the global model $G$ and the local model $L_k$.

- **Uniform Distribution:** When $\alpha = 1$, the Beta distribution becomes uniform over [0,1]. In this case, $\lambda$ is sampled uniformly across the entire interval, meaning that each model, $G$ and $L_k$, has an equal probability of being weighted more or less in the mixup process. This leads to a broad exploration of different combinations of global and local models, allowing for a wide range of mixed models.

- **Concentrated Mixup ( $\alpha > 1$):** When $\alpha > 1$, the Beta distribution is concentrated around the center of the interval [0,1]. As a result, the mixup factor $\lambda$ is more likely to be closer to 0.5, leading to more balanced combinations of the global and local models. This results in outputs that are more "mixed," with neither model dominating the mixup process. Such a setup can enhance the robustness of the combined model, as it prevents extreme weighting of either model, creating smoother interpolations between them.

- **Extremal Mixup ( $\alpha < 1$):** In contrast, when $\alpha < 1$, the Beta distribution becomes U-shaped, with more probability mass near 0 and 1. This means that $\lambda$ tends to be either very close to 0 or very close to 1, favoring one model over the other in the mixup process. When $\lambda \approx 0$, the local model $L_k$ is chosen almost exclusively, and when $\lambda \approx 1$, the global model $G$ is predominantly selected. This form of mixup creates a more deterministic selection between global and local models, with less mixing occurring.

---

[8]Classification accuracy using softmax

Table 5: Performance (%) on Caltech-101 (CNN backbone). Rows A–D denote evaluation angles; AVG is the mean across A–D.

| $N = 100,\ C = 100\%$ | | | | | |
|---|---|---|---|---|---|
| **Method** | A | B | C | D | AVG |
| pMixFed (Ours) | 78.4 | 80.2 | 79.1 | 79.5 | **79.3** |
| FedRep | 75.8 | 78.5 | 77.2 | 76.6 | 77.0 |
| Per-FedAvg | 75.1 | 77.2 | 76.0 | 77.5 | 76.5 |
| FedAlt | 70.8 | 74.1 | 73.5 | 73.9 | 73.1 |
| $N = 10,\ C = 100\%$ | | | | | |
| **Method** | A | B | C | D | AVG |
| pMixFed (Ours) | 80.9 | 83.8 | 82.9 | 81.7 | **82.4** |
| FedRep | 78.6 | 80.9 | 80.4 | 80.5 | 80.1 |
| Per-FedAvg | 79.0 | 81.8 | 82.1 | 81.0 | 81.0 |
| FedAlt | 74.0 | 76.8 | 75.2 | 76.0 | 75.5 |
| $N = 100,\ C = 10\%$ | | | | | |
| **Method** | A | B | C | D | AVG |
| pMixFed (Ours) | 74.6 | 77.8 | 76.9 | 74.9 | **76.1** |
| FedRep | 72.9 | 74.6 | 74.5 | 74.8 | 74.2 |
| Per-FedAvg | 72.1 | 74.0 | 73.8 | 74.1 | 73.5 |
| FedAlt | 67.4 | 70.8 | 69.9 | 69.8 | 69.5 |
| $N = 10,\ C = 10\%$ | | | | | |
| **Method** | A | B | C | D | AVG |
| pMixFed (Ours) | 77.2 | 80.1 | 79.8 | 79.0 | **79.0** |
| FedRep | 75.1 | 77.3 | 77.2 | 77.8 | 76.9 |
| Per-FedAvg | 74.2 | 76.5 | 76.0 | 77.9 | 76.2 |
| FedAlt | 70.5 | 72.6 | 72.9 | 72.8 | 72.2 |

The behavior of different $\alpha$ values is depicted in Figure 9, where the distribution of the mixup factor $\lambda$ is visualized. These distributions highlight the varying degrees of mixup, ranging from uniform blending to nearly deterministic model selection. To thoroughly investigate the impact of $\alpha$ on model performance, we designed two distinct experimental setups:

- **Random Sampling:** In this scenario, we set different $\alpha$, meaning that the $\lambda$ values are sampled uniformly from the interval [0,1] according. This ensures a wide range of mixup combinations between the global and local models. The random sampling approach helps us assess the general robustness of the model when the mixup degree $\lambda$ is not biased towards any specific value. Table 6 shows the effect of different $\alpha$ on the overall test accuracy of *pMixFed*.

- **Adaptive Sampling:** For this case, we divided the communication rounds into three distinct stages, each consisting of $\frac{epoch_{global}}{3}$ epochs. During these stages, the parameter $\alpha$ is adaptively changed as

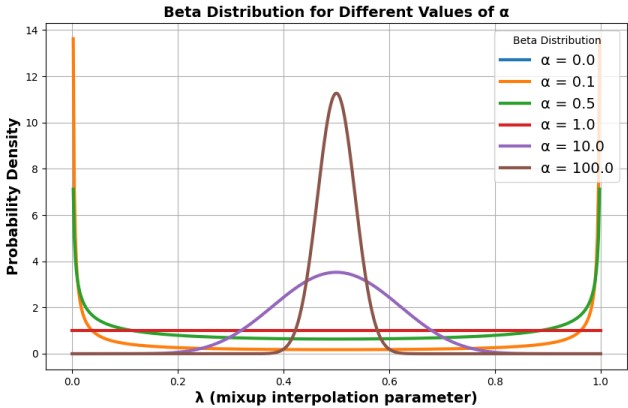

Figure 9: PDF $\lambda$ (mixup degree) for different values of $\alpha$ in $\beta$ distribution

follows:

$$\alpha = \begin{cases} 0.1, & \text{initial stage (early training)} \\ 100, & \text{middle stage (convergence phase)} \\ 10, & \text{final stage (fine-tuning)} \end{cases} \tag{14}$$

This adaptive strategy mimics the behavior of the original pFedMix algorithm while also allowing for more controlled exploration of different mixup combinations. During the early and late stages, a small $\alpha$ (0.1) encourages more deterministic model selections (i.e., either local or global), while the middle stage with $\alpha = 100$ promotes more balanced mixing. This dynamic adjustment of $\alpha$ enables us to control the degree of mixup at different phases of training. Table 7 also shows the effect of different sampling approaches (random and adaptive) on the overall test accuracy of all three dataset *pMixFed*. The results shows that the adaptive sampling which creats a form scheduling for mixup degree shows promising result even compared to the original algorithm using adaptive $\mu$.

Table 6: Accuracy of $pMixFed$ with Different $\alpha$ Values in the Beta Distribution

| Dataset | $\alpha = 0.1$ | $\alpha = 0.5$ | $\alpha = 1$ | $\alpha = 2$ | $\alpha = 5$ |
|---|---|---|---|---|---|
| CIFAR-10 | 78.5 | 81.2 | 82.6 | 84.3 | 83.1 |
| CIFAR-100 | 42.3 | 45.6 | 47.2 | 49.8 | 48.1 |

Table 7: Accuracy of $pMixFed$ with Different $\alpha$ Values and Sampling Strategies

| Dataset | Sampling Strategy | Accuracy(%) |
|---|---|---|
| CIFAR-10 | Random Sampling ($\alpha = 1$) | 82.6 |
| CIFAR-10 | Adaptive Sampling | 85.3 |
| CIFAR-100 | Random Sampling ($\alpha = 1$) | 47.2 |
| CIFAR-100 | Adaptive Sampling | 50.1 |
| MNIST | Random Sampling ($\alpha = 1$) | 97.9 |
| MNIST | Adaptive Sampling | 98.6 |

### E.6 Communication and Computational Cost

Although *pMixFed* introduces additional computation compared to partial personalization baselines (e.g., FedAlt, FedSim), we analyze its efficiency in terms of communication and runtime. Figure 10(a) shows the number of layers frozen at each communication round due to a zero mixup degree, which naturally reduces

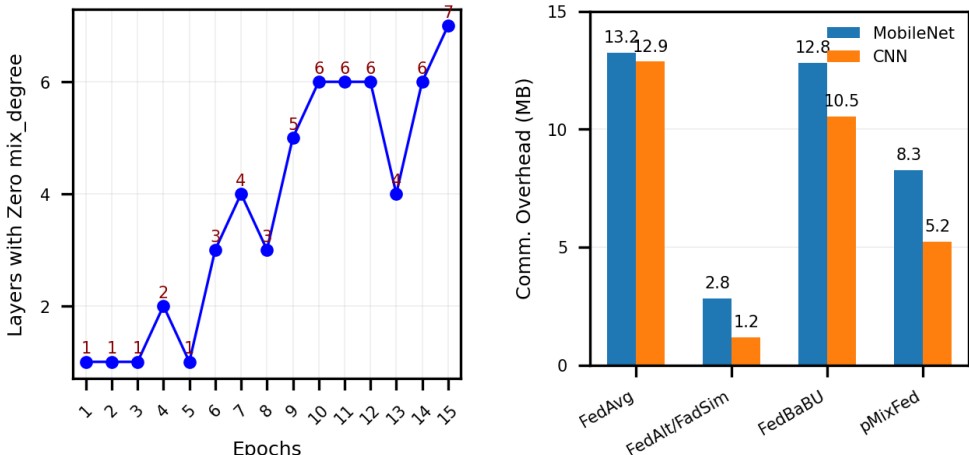

Figure 10: (a) Number of frozen layers per communication round due to zero mixup degree. (b) Average communication overhead per round across 100 clients for CNN and MobileNet on CIFAR-100.

the amount of information exchanged. Figure 10(b) reports the average communication overhead across 100 clients for CNN and MobileNet on CIFAR-100.

Compared to full-model FL (FedAvg), *pMixFed* does not increase the parameter size transmitted per layer. While its per-round communication cost is higher than partial methods (FedAlt, FedSim), two key properties compensate for this overhead:

**Faster convergence and early stopping.** *pMixFed* reaches target accuracy in significantly fewer communication rounds, reducing the overall cost. In large-scale and highly heterogeneous settings (e.g., MobileNet on CIFAR-100), FedAlt and FedSim often diverge or require far more rounds to achieve comparable accuracy.

**Adaptive layer freezing.** The mixup degree $\lambda_i^{(t)}$, updated dynamically via Eq. (5)–(6), drives progressive freezing of layers. As training advances, an increasing number of layers are excluded from both broadcasting and aggregation, further reducing transferred parameters in later rounds. We also evaluated computational efficiency by measuring GPU usage and runtime per round (Table 8). In the most demanding case (CIFAR-100, MobileNet, $C = 100\%$), *pMixFed* is only slightly slower (1–7s per round) and requires marginally more GPU memory (1–22MB). Given its faster convergence (on average, $\sim$10% fewer rounds) and improved stability, the effective overhead is substantially mitigated.

Table 8: GPU usage and computational time per global round using MobileNet on CIFAR-100 ($C = 100\%$).

| Method | $N = 100$ | | $N = 10$ | |
|---|---|---|---|---|
| | Time (s) | Memory (MB) | Time (s) | Memory (MB) |
| FedAvg | 94.16 | 5362.44 | 95.31 | 735.25 |
| FedAlt | 94.75 | 5372.77 | 92.86 | 746.85 |
| FedSim | 95.72 | 5378.19 | 91.43 | 746.85 |
| **pMixFed** | **100.03** | **5376.90** | **98.48** | **762.99** |

*In summary, while pMixFed incurs a minor per-round overhead, its faster convergence and adaptive freezing mechanism reduce the overall communication and computation cost, making it more scalable and robust in heterogeneous FL settings.*

