# OpenReview forum: "pMixFed: Efficient Personalized Federated Learning through Adaptive Layer-Wise Mixup"
_TMLR — Withdrawn by Authors_

### Review · Reviewer_2PYq · 2026-04-18

**Summary Of Contributions:**

This paper introduces pMixFed, a personalized federated learning approach that adaptively mixes global shared layers and local personalized layers to improve model performance. Unlike existing PFL approaches, pMixFed does not require a hard split between shared and personalized layers, while also allowing for the flexibility to adapt the degree of personalization across the model depth. Furthermore, the authors propose to adjust the mixing factor per round based on the relative accuracy of the client in each round. Evaluations are conducted on four datasets with varying heterogeneity levels, client participation rate and population size.

**Audience:**

No

**Audience Explanation:**

Given the significant issues with the paper's methodology, presentation, and empirical evaluation, it is unlikely that many individuals in TMLR's audience would find the findings of this paper to be valuable in its current form. While the problem and solution proposed by the paper are relevant and interesting, the paper requires significant revisions before it can be considered for publication.

**Claims And Evidence:**

No

**Claims Explanation:**

The paper has several flaws that undermine the credibility of the claims made. I am enlisting them below:

1. Significant notational inconsistencies and errors: Throughout the paper, there are numerous notational inconsistencies and errors that make it difficult to follow the core methodology.
    - Eq. (2) is likely incorrect. Why is there a $1/|\mathcal{D}_k|$ term when $\mathcal{F}_k$ is itself defined as an average over the dataset?
    - Eq. (6) is recursively defined in terms of itself, which is not mathematically valid.
    - Below eq. (5), all local test sets are denoted as $x = \\{x_1, x_2, \ldots, x_K\\}$ while $x$ is repeatedly used in the paper as a single data point.
    - Below eq. (8), the authors use $LM_i^{(0)}$ and $GM^{(0)}$ to suddenly refer to the size of the models. Why is the superscript (0) used here and what does the indexing over $i$ in $LM$ refer to – clients or layers?
    - E1. (1) should have $(x_i, y_i) \in \mathcal{D}_k$ instead of $(x, y_i) \in \mathcal{D}_k$.
    - There are many more instances of confusing notations such as the ones surrounding $\xrightarrow{localupdate}$. This is a highly non-standard usage.
    - Eq (3.1) is never defined but referenced in the text.

2. It is unclear what the authors mean by *"It is important to note that the sizes of local models $LM_i^{(0)}$ can differ from each other. Consequently, the size of the global model $GM^{(0)}$ should be greater than the maximum size of local models."* How does the aggregation work when local models have different sizes? How is the global model size determined then?

3. The discussion of the most important component of the algorithm, the mixing factor $\mu$, is deferred to appendix with no intuition or insight being provided in the main text. This is a significant flaw in the presentation of the paper as $\mu$ is central to the algorithm and should be explained clearly in the main text.

4. Several issues with empirical evaluation:
    - In Figure 3, the evaluated baselines seem to not learn anything, likely due to the an incorrect hyperparameter setting or bug in the code. I am surprised by the accuracy of FedAvg staying around 25\% in Table 1 for CIFAR-10 when the evaluted heterogeneity seems fairly modest while pMixFed achieves 69\%. The choice of running the algorithms for only T=50 global rounds is also clearly unjustified as the baselines do not seem any close to convergence.
    - Section 5.1.3 mentions Adam optimizer is being used but does not specify for which algorithm. In FL, Adam is not commonly used on the client side due to its high memory requirements. Thus, it is important to clarify which algorithms are using it and how it is being applied.
    - The heterogeneous data partitions seem very modest, giving 50% well balanced classes to each client. The algorithm should be tested on more challenging heterogeneity settings such as with Dirichlet partitioning (e.g. see Flower framework).
    - Ablation studies in Section 5.4 lack discussion and depth. Results in Figure 5(a) remain quite unclear and corresponding text provides no explanation. From Figure 5(b), it once again seems that hyperparameters settings are incorrect even for pMixFed as the accuracy rises in first 10 rounds and then continues to drop.
    - In Section 5.3, the authors discuss pitfalls of baselines when new users are added into the cohort. However, the setup of this experiment is not clearly described. How are new users added? What data do they have?
    - Figure 4 reports 100% accuracy for pMixFed. This seems highly unrealistic.

**Requested Changes:**

1. The paper needs to be thoroughly revised to address the notational inconsistencies and errors. The authors should ensure that all equations are mathematically valid.
2. The authors should provide a clear and intuitive explanation of the mixing factor $\mu$ in the main text, along with insights into how it is determined and its impact on the algorithm's performance.
3. The empirical evaluation needs to be significantly improved and checked for correctness. It is surprising that the well established baselines have 20-30% lower accuracy than pMixFed.

---

### Review · Reviewer_jm3q · 2026-05-01

**Summary Of Contributions:**

This paper proposes pMixFed, a partial personalized federated learning (PFL) method that replaces a static, hard model split with adaptive, layer-wise interpolation (“mixup”) between local and global parameters during both broadcasting and aggregation. The core idea is to compute per-layer mixing coefficients that vary across depth and are adapted per round and per client based on relative performance, aiming to mitigate global–local discrepancy, client drift, and catastrophic forgetting. Experiments on MNIST, CIFAR-10/100, and Caltech-101 report improved average accuracy and smoother convergence compared to several PFL and FL baselines.

Strengths:

1.	The paper targets an important issue in partial personalized FL: fixed layer partitioning is often too rigid under heterogeneous client distributions. The discussion around global-local discrepancy, client drift, cold-start clients, and catastrophic forgetting is meaningful and relevant to the PFL literature.

2.	The proposed layer-wise interpolation is intuitive, easy to implement, and addresses a real limitation of hard layer partitioning.

3.	Applying a mixup-like idea in parameter space for PFL (in both broadcast and aggregation) is an interesting design that differs from prior data-space mixup approaches in FL.

4.	The paper also includes useful robustness and efficiency analyses beyond standard accuracy comparisons.

Weakness

1.	The evaluation may involve test leakage. The paper appears to use test accuracy to adapt the mix factor and also states that models are fine-tuned on test data during evaluation at least for FedAlt.

2.	Whether all methods are evaluated under the same stopping/evaluation protocol? Reporting “the highest accuracy achieved” across rounds due to training drops is cherry-picking and undermines fair comparison. I think standard practice is to report early-stopping accuracy with validation set for all methods consistently.

3.	The paper should include explicit heterogeneous-architecture experiments if it wants to claim model heterogeneity support.

4.	The empirical comparison could be strengthened with more recent and closely related baselines. Most current baselines are standard or relatively old FL/PFL methods, such as FedAvg, Ditto, and FedSim. Given that adaptive personalization and layer-wise aggregation have been explored in recent FL literature, the paper should include stronger recent baselines such as FLAYER [1] and FedLWS  [2]. This would make the advantage more convincing.

5.	The introduction is not always self-contained enough. Some claims are stated as if they are obvious, while the reasoning behind them is not fully explained. For example, the paper states that when a client’s performance drops due to new incoming data, it “requires more frozen layers.” This is not self-evident, since under distribution shift, the client may need more adaptable local layers instead of freezing more layers. Similarly, the statement that “frozen local model updates can diverge” is confusing, because frozen layers should not be updated. Overall, I think the introduction would be stronger if the authors made the reasoning more rigorous.

6.	The paper has some inconsistent terminology, and formatting problems, which make the method harder to follow. For example, t the beginning of Section 4, the paper says the method incorporates layer-wise Mixup in the feature space, but Section 4.2 later says pMixFed applies Mixup in the model parameter space rather than the feature space. The interpretation of λ is also confusing: one paragraph states that λ=1 means 100% global sharing and λ=0 means the local layer is frozen, while another paragraph states that λ=0 for the base layer indicates total sharing and λ=1 for the final layer means no sharing.

[1] Optimizing Personalized Federated Learning through Adaptive Layer-Wise Learning.

[2] FedLWS: Federated Learning with Adaptive Layer-wise Weight Shrinking.

**Audience:**

Yes

**Audience Explanation:**

The topic is relevant to TMLR because personalized federated learning is an active and important area. The idea of adaptive layer-wise mixing between global and local models could be interesting to FL/PFL researchers.

**Broader Impact Concerns:**

I don’t see major ethical or broader-impact concerns specific to this work.

**Claims And Evidence:**

No

**Claims Explanation:**

Please see the detailed weaknesses above. Together, these issues make the empirical and methodological claims less fully supported.

**Requested Changes:**

Could the authors please either clarify or address my concerns about the weaknesses listed above?

---

### Review · Reviewer_YVGJ · 2026-05-08

**Summary Of Contributions:**

The paper proposes **pMixFed**, a personalized federated learning (PFL) framework that replaces the conventional hard partition between shared and personalized layers with an adaptive layer-wise mixing strategy. Instead of statically deciding which layers are global or local, the method gradually interpolates between global and local parameters using a Mixup-inspired mechanism directly in parameter space.

### Strengths
- The paper studies an important problem in personalized federated learning, namely the limitations of static layer partitioning.
- The idea of gradually mixing local and global parameters across layers is interesting and differs from existing hard-splitting approaches.
- The experimental section covers several datasets and personalization baselines.

### Weaknesses (details after)

- Several core algorithmic details are difficult to follow, especially regarding the definition and update rules of the adaptive mixing coefficient $\mu$ and its relationship to $\lambda$.
- The descriptions of the broadcasting and aggregation procedures are sometimes inconsistent with the algorithms and equations provided.
- Important implementation details are missing, particularly regarding how accuracies are computed and communicated during aggregation.
- Some statements appear contradictory or ambiguous (e.g., whether only one client is mixed with the global model in each round).
- The presentation contains a noticeable number of typos, notation inconsistencies, and unclear explanations, which significantly affect readability.
- While the empirical results are promising, the theoretical motivation and justification for some design choices remain underdeveloped.

**Audience:**

Yes

**Audience Explanation:**

The topic interests the community of Federated Learning

**Broader Impact Concerns:**

I do not have major ethical concerns regarding the proposed method.

**Claims And Evidence:**

No

**Claims Explanation:**

## Main Concerns

### 1. Difficulty understanding the broadcasting stage

The description of how $\mu_k^t$ in Equation (5) is computed is unclear and somewhat confusing.

- In particular, the meaning of $Acc_k^t$ and $Acc_{overall}^{t-1}$ during the broadcasting phase is not well defined.
- The sentence introducing the computation of `Acc` is difficult to parse and should be rewritten more carefully.

I am also confused about the relationship between Equation (5) and Algorithm 1:

- In Algorithm 1, line 7 uses $\mu_{\text{broad}}$, but this quantity is never explicitly defined.
- How is $\mu_{\text{broad}}$ related to the $\mu_k^t$ updated in line 4?
- Is $\mu_{\text{broad}}$ client-specific or shared across all clients?

These details are important because the adaptive behavior of the method depends heavily on this coefficient.

---

### 2. Confusion regarding the aggregation stage

Section 4.2.2 states that $\mu$ is constant for all clients during aggregation. However, this seems inconsistent with Algorithm 2.

- If $\mu$ is identical for all clients during aggregation, why does Algorithm 2 line 4 update $\mu_k^t$ for every client individually?
- The implementation details of the aggregation-stage accuracy computation are also unclear.

Specifically:

- How is the “test accuracy of the previous global model on all local test sets” obtained?
- Is the evaluation performed locally on each client and then transmitted to the server?
- Or does the server have access to a centralized validation/test dataset?

This distinction is important for both privacy assumptions and implementation feasibility.

In addition:

- If $\mu$ is constant across clients during aggregation, then $\lambda_i$ is also constant across clients according to Equation (6).
- In that case, the term $\lambda_{i,k}^{(t)} \cdot G_i^{(t)}$ in Equation (8) could seemingly be computed directly by the server without requiring client-specific processing.

This part of the method would benefit from a clearer and more rigorous explanation.

---

### 3. Presentation of the $\lambda_i$ formulation

I found the presentation of Equation (6) somewhat difficult to follow.

The equations could be written more clearly as:

- **Broadcasting stage**

  $$
  \lambda_i = \min\{1, \mu (n-i)\}
  $$

- **Aggregation stage**

  $$
  \lambda_i = \max\{0, 1 - i\mu\}
  $$

This formulation may improve readability and avoid the current piecewise notation, which is harder to parse.

---

### 4. Repeated and potentially contradictory explanation on page 8

At the end of page 8, there are two very similar paragraphs:

- one referring to Algorithm 1,
- and another referring to Algorithm 2.

Both appear to repeat essentially the same argument, making it unclear which statement applies to which algorithm.

Moreover, the text states:

> “only one client is ‘Mixed up’ with the global model”

This statement is confusing because:

- it was never emphasized earlier in either Algorithm 1 or Algorithm 2,
- and it seems inconsistent with the overall federated learning setup where multiple participating clients are processed in each round.

This point should be clarified carefully.

---

### Typos and Minor Issues

1. Section 3.1:

   > “solving Equation 3.1”

   should be:

   > “solving Equation 3”

2. Section 4.2, page 6:

   > “Appendix Sec. 2”

   should likely be:

   > “Appendix B”

3. Page 8:

   > “Accordingly, A history”

   should be:

   > “Accordingly, a history”

4. Section 4.2.2, page 8:

   > “In this scenario, $\lambda = 0$ for the first base layer”

   According to Equation (6), shouldn’t this instead be:

   > $\lambda = 1$ for the first base layer?

**Requested Changes:**

I would suggest that the authors carefully revise Section 4.2 to provide a clearer and more consistent description of the proposed algorithm. In its current form, several parts of the broadcasting and aggregation procedures are difficult to follow, and some notations and explanations appear ambiguous or inconsistent with the algorithms and equations.

---

### Comment · Action_Editor_rEtC · 2026-05-19
**Author/reviewer discussion**

Hello all. I just want to remind the authors to please engage with the reviews, as they are all complete now. Please use this time to respond to things brought up by the reviewers, either about ways that issues might already be addressed in the work, or some discussion of how easy they would be to do in a final version of the paper.

---

### Note · Authors · 2026-05-29

**Comment:**

Dear Action Editor and Reviewers,

Thank you very much for the thoughtful and detailed feedback on our submission. We sincerely appreciate the time and effort invested in reviewing our work and providing constructive suggestions.

After carefully studying the reviews, we recognize several important areas where the paper requires substantial improvement. In particular, we acknowledge the concerns regarding the clarity of the methodological presentation, consistency of notation and definitions, experimental rigor, and the positioning and explanation of our contributions. We also appreciate the reviewers’ detailed observations about the evaluation protocol, baseline comparisons, algorithmic description, and theoretical presentation. Given the scope of the requested revisions, we have decided to withdraw the current submission and work on a substantially revised version that more thoroughly addresses the concerns raised. We plan to improve the clarity of the methodology, strengthen the empirical evaluation, refine the theoretical motivation and presentation, and incorporate additional analyses and comparisons where appropriate.

We greatly value the reviewers’ feedback and believe these revisions will significantly strengthen the work. We hope to resubmit a substantially improved version in the coming weeks.

Thank you again for your time and valuable suggestions.

**Withdrawal Confirmation:**

I have read and agree with the venue's withdrawal policy on behalf of myself and my co-authors.